# Unique structure of active platinum-bismuth site for oxidation of carbon monoxide

Bing Nan[1,2], Qiang Fu [3], Jing Yu[4], Miao Shu[1], Lu-Lu Zhou[3], Jinying Li[3], Wei-Wei Wang[3], Chun-Jiang Jia [3✉], Chao Ma [5✉], Jun-Xiang Chen[6], Lina Li[1,7] & Rui Si [1,7✉]

As the technology development, the future advanced combustion engines must be designed to perform at a low temperature. Thus, it is a great challenge to synthesize high active and stable catalysts to resolve exhaust below 100 °C. Here, we report that bismuth as a dopant is added to form platinum-bismuth cluster on silica for CO oxidation. The highly reducible oxygen species provided by surface metal-oxide (M-O) interface could be activated by CO at low temperature (~50 °C) with a high $CO_2$ production rate of 487 $\mu mol_{CO2} \cdot g_{Pt}^{-1} \cdot s^{-1}$ at 110 °C. Experiment data combined with density functional calculation (DFT) results demonstrate that Pt cluster with surface Pt−O−Bi structure is the active site for CO oxidation via providing moderate CO adsorption and activating CO molecules with electron transformation between platinum atom and carbon monoxide. These findings provide a unique and general approach towards design of potential excellent performance catalysts for redox reaction.

[1] Shanghai Institute of Applied Physics, Chinese Academy of Sciences, Shanghai, China. [2] University of Chinese Academy of Science, Beijing, China. [3] Key Laboratory for Colloid and Interface Chemistry, Key Laboratory of Special Aggregated Materials, School of Chemistry and Chemical Engineering, Shandong University, Jinan, China. [4] Shanghai Institute of Measurement and Testing Technology, Shanghai, China. [5] Center for High Resolution Electron Microscopy, College of Materials Science and Engineering, Hunan University, Changsha, China. [6] Division of China, TILON Group Technology Limited, Shanghai, China. [7] Shanghai Synchrotron Radiation Facility, Zhangjiang Laboratory, Shanghai Advanced Research Institute, Shanghai, China. ✉email: jiacj@sdu.edu.cn; cma@hnu.edu.cn; sirui@sinap.ac.cn

The CO oxidation reaction ($CO + 1/2\ O_2 = CO_2$) is a well-known model reaction in heterogeneous catalysis, as well as a key step to resolve automobile exhaust containing CO, NO, and hydrocarbons[1–5]. According to the previous reports, the platinum (Pt)-based catalysts exhibit excellent catalytic activity in CO oxidation. In one hand, these high-performance catalysts always require reducible oxides as supports, such as $CeO_2$[2], $FeO_x$[3], $MnO_2$[4], and $Co_3O_4$[4], due to their rich surface oxygen vacancy and the so-called "strong metal-support interaction"[6–8]. In another hand, the irreducible oxide ($SiO_2$ and $Al_2O_3$)-supported platinum catalysts with the advantages of commercial production, low cost, and extensive application frequently show poor catalytic activity in CO oxidation especially in low temperature because of the lack of surface activated oxygen and suitable active site[9,10]. Meanwhile, the aggregation of platinum species in $SiO_2$-supported platinum catalysts frequently results in the deactivation of catalysts in CO oxidation. Therefore, it is a big challenge to develop a kind of irreducible oxide-supported platinum catalyst with both excellent catalytic performance in low temperature and thermo-stability for practical application.

Many research groups have found that the addition of a secondary element, such as Sn[5], K[11], Co[12], and Bi[13,14], distinctly improved the activity of silica- or alumina-supported platinum catalysts, no matter in the form of oxide clusters or metallic alloy for Pt. As for bismuth element, it has been widely used as the secondary element to improve the catalytic activity in various oxidative reactions[13,14], due to providing high content of mobile oxygen[13], preventing the overoxidation of noble metal[15] and suppressing adsorption of poisoning species[16]. Many research groups have prepared crystalline platinum-bismuth alloy to promote the activity of some oxidation reactions. However, bismuth as an oxyphilic element is easily to form $Bi^{3+}$ species in oxidative atmosphere resulting in oxidation of platinum-bismuth alloy. Feng et al.[17] reported that the metallic and positive bismuth species coexists in Pt-Bi/SBA-15 catalysts for the selective oxidation of glycerol. In addition, the deposit location of bismuth species (on the surface of support or active site) also have huge influence on catalytic performance[17,18]. Therefore, it is significant for us to determine the precise local structure of active site (alloy or oxide cluster?[14]) and the practical valance state of bismuth and platinum (metallic or positive charge?[19,20]) for Bi-promoted platinum catalysts in oxidation reactions. In this work, we provide a special viewpoint of active site: partially oxidized Pt–[O]$_x$–Bi structure totally different with PtBi alloy to improve the thermo-stability of catalysts and supply active oxygen species for efficiently catalyzing CO oxidation at low temperature.

Moreover, many researchers have identified that various interfaces in platinum-based catalysts play a key role in many industrially important reactions, such as metal−support[3,21,22], metal−oxide[23–25] and metal−metal hydroxide[26]. Chen et al. reported iron nickel hydroxide-platinum nanoparticles (Pt–OH–Fe/Ni) were highly efficient for CO oxidation owing to abundant sites of Pt–OH–M interfaces[26]. According to previous reports[27,28], it is easy to build multifarious atomic interface between metal and reducible oxide to improve catalytic performance. However, it is quite difficult to build effective metal-support or metal-oxide interface due to the infertile oxygen, poor reducibility and over-stable surface composition of irreducible oxide ($SiO_2$ and $Al_2O_3$). Therefore, it is great research interests to build stable and efficient interfaces via the doping of secondary element on inert support to improve the catalytic performance in CO oxidation.

In this work, we prepared silica-supported platinum-bismuth catalysts via an incipient wetness impregnation, possessing excellent sinter resistance due to the formation of oxidized $Pt_xBi_yO_z$ cluster. The dopant of bismuth species builds a unique interface between platinum and bismuth (Pt−[O]$_x$−Bi structure) species, which exhibits an absolutely different active site compared with pure platinum sample, providing superior active oxygen species activated by CO at low temperature (~50 °C) with a high $CO_2$ production rate of 487 $\mu mol_{CO2}\ g_{Pt}^{-1}\ s^{-1}$ at 110 °C. Meanwhile, this interface structure prevents oversaturated CO adsorption from poison of platinum species and activates CO molecules to catalyze CO oxidation in a lower apparent activation energy.

## Results

**Structural characterization of bismuth-doped platinum samples**. A serial of silica-supported platinum and platinum-bismuth samples were prepared by a co-incipient wetness impregnation method. The bulk concentrations of platinum and bismuth (Pt: 0.8, 0.9, and 0.9 wt.%; Bi: 2.3, and 6.1 wt.% for 1Pt–SiO2, 1Pt2Bi–SiO2, 1Pt5Bi–SiO2 respectively) are close to these designed numbers, indicating the disposition of bismuth species have no effect on the loading of platinum (Supplementary Table 1). Furthermore, the Bi-free and Bi-doped samples have similar textural properties, such as $S_{BET}$ values and the type of adsorption–desorption isotherms (Supplementary Table 1 and Supplementary Fig. 1), in which we can exclude the physical effect on the following catalytic performance.

Small-size oxide species were quite stable on silica surface in Bi-promoted samples in aberration-corrected high-angle annular dark-field imaging scanning transmission electron microscopy (HAADF-STEM) images (Fig. 1a, b), even after heat treatment in air. Only ultrafine clusters of $1.7 \pm 0.4$ nm with narrow size-distribution were created in 1Pt2Bi-SiO2 (Supplementary Fig. 2e, f and 3a), without any crystallized platinum or bismuth metal/oxide particles. However, after calcination in air, huge metallic Pt particles of ~100 nm (Supplementary Fig. 2a) and platinum oxide clusters ($1.6 \pm 0.5$ nm) simultaneously appeared in 1Pt–SiO2 (Supplementary Fig. 2b, c and 3b). It illustrates that the addition of bismuth element could suppress the growth of metal/oxide particles, similar to the promotion by alkali ions[11] and silica support shows poor ability to stabilize platinum species. Furthermore, the corresponding aberration-corrected energy dispersive spectroscopy (EDS) mapping results of 1Pt2Bi–SiO2 (Fig. 1c) show that platinum and bismuth elements distribute uniformly at the cluster level within the same areas on the surface of $SiO_2$ without evident separation. When the EDS mapping was conducted for the individual cluster, no obvious core-shell structure can be observed (Supplementary Fig. 3c). Xie et al. also reported that bismuth species was deposited selectively on the Pt particles rather than carbon support[29]. Meanwhile, the X-ray diffraction (XRD) also confirmed the promotion of bismuth species, in which no obvious $Pt/PtO/PtO_2/Bi/Bi_2O_3$ phase was detected in 1Pt2Bi–SiO2, even in a "slow-scan" mode (Fig. 1d and Supplementary Fig. 4). When bismuth oxide species were deposited on silica separately (2Bi-SiO2), it also stabilized in small-size without any diffraction peaks of $Bi/Bi_2O_3$ in XRD profiles (Fig. 1d). However, the observation of sharp Pt peaks (39.7° and 46.2°) verifies the formation of huge Pt particle in 1Pt-SiO2. As shown in Fig. 1c, the dopant of bismuth species reaches optimization at 2 wt.% and the formation of broad diffraction peak of $Bi_2O_3$ for 1Pt5Bi-SiO2. Therefore, the HAADF-STEM and XRD results directly indicated that bismuth oxide species as a promoter could phenomenologically prevent the formation of pure huge Pt particles (50−100 nm) on an inert support ($SiO_2$), which is further supported by our density functional theory calculations with more details in the Supplementary Table 2.

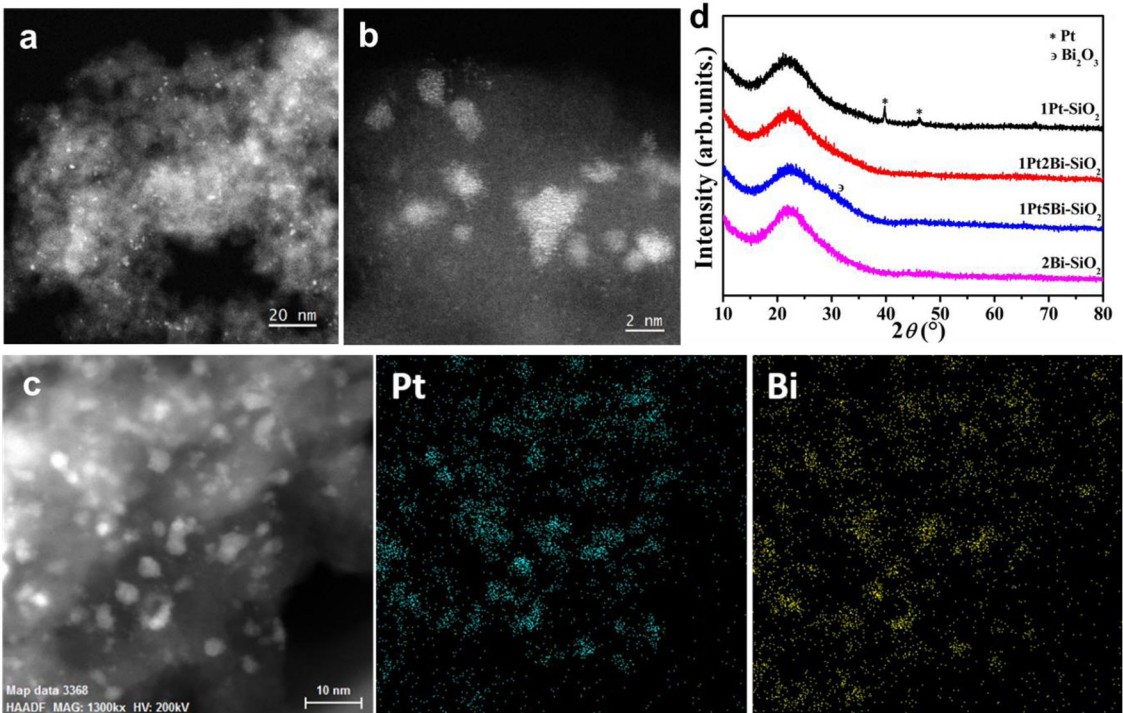

**Fig. 1 Structural characterization of Pt/PtBi–SiO₂ samples.** Representative aberration-corrected HAADF-STEM images (**a**, **b**) at different scales and corresponding STEM-EDS elemental mapping images **c** for fresh 1Pt2Bi-SiO₂; XRD patterns (**d**) of fresh Pt/PtBi-SiO₂ samples.

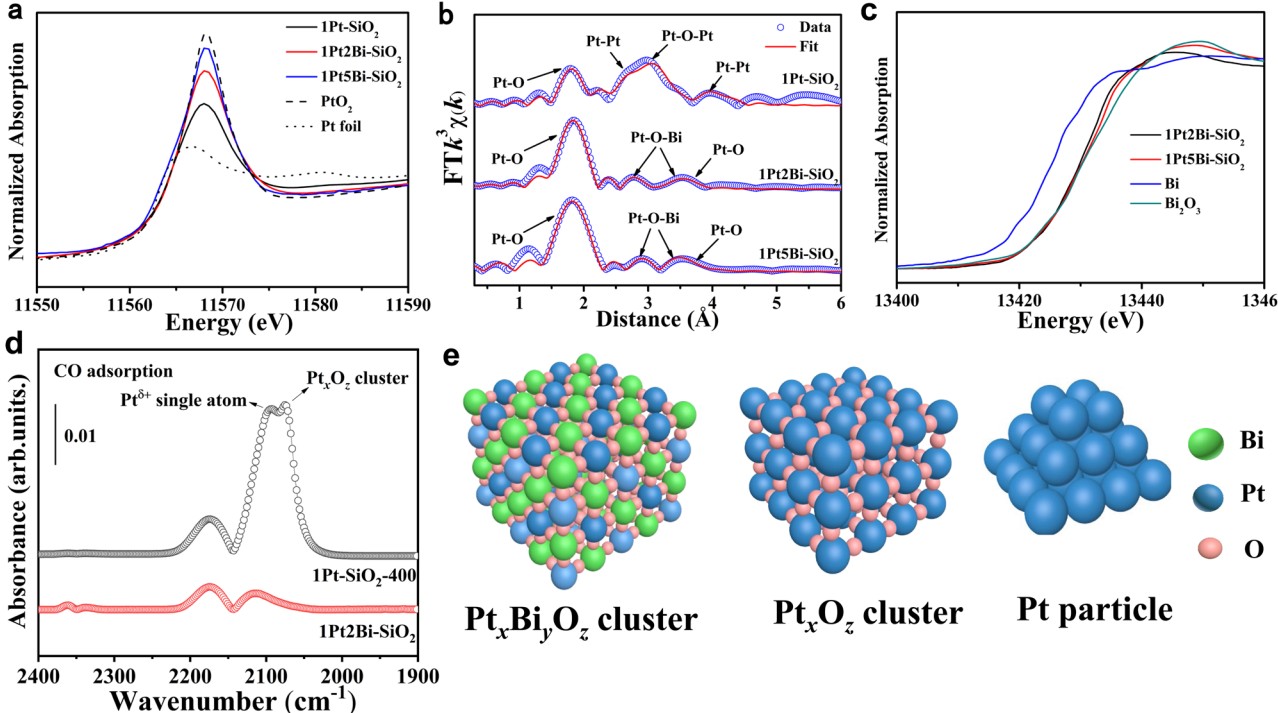

**Fig. 2 Local coordination structure of Pt/PtBi–SiO₂ samples.** Pt $L_3$–edge **a** XANES profiles, **b** EXAFS fitting results in $R$ space and Bi $L_3$–edge **c** XANES profiles of Pt/PtBi–SiO₂ samples; **d** in-situ DRIFTS experiments for CO adsorption of 1Pt2Bi-SiO₂ and 1Pt-SiO₂-400; **e** Schematic demonstration on platinum-bismuth oxide clusters, platinum oxide clusters, and metallic Pt particles.

**The local coordination structure of platinum and bismuth species.** According to aberration-corrected HAADF-STEM images, we can observe the existence of platinum-bismuth oxide clusters. X-ray absorption fine structure (XAFS) technique was applied to clarify the local structure of this cluster in Bi-promoted samples. The X-ray absorption near-edge structure (XANES) of Pt–$L_3$ edge profiles (Fig. 2a and Supplementary Table 3) showed platinum is in a low valence state (+1.8), due to the formation of

huge metallic Pt particles in 1Pt-SiO$_2$. As the increasing of bismuth-dopant, the average oxidation state of platinum arises from 1.8 to 3.5, because bismuth oxide species could make interaction with platinum to stabilize abundant oxygen around Pt atom and thus enhancing average valance state of platinum. The extended X-ray absorption fine structure (EXAFS) fitting results (Fig. 2b and Supplementary Table 2) indicated that the major Pt−O ($R \approx$ 2.0 Å, $CN \approx 5$) shell plus an apparent Pt−Bi ($R \approx 3.0$ Å, $CN \approx 4$) component originated from the Pt−[O]$_x$−Bi structure, further demonstrating the formation of homogeneous oxidized Pt$_x$Bi$_y$O$_z$ cluster (Fig. 2e). Meanwhile, the XANES profiles of Bi−L$_3$ edge (Fig. 2c) and X-ray photoelectron spectroscopy (XPS) spectra display that bismuth species were in the pure Bi$^{3+}$ state for Bi-promoted samples (Supplementary Fig. 5), which could exclude the possibility of forming PtBi alloy. While for 1Pt–SiO$_2$, without strong interaction between Pt and SiO$_2$, a dominant Pt−Pt metallic shell ($R \approx 2.8$ Å, $CN \approx 6.4$) from metallic Pt particle and a minor Pt−O−Pt shell ($R \approx 3.1$ Å, $CN \approx 2.7$) from small-size Pt$_x$O$_z$ cluster could be determined. In addition, in-situ diffuse reflectance infrared Fourier transform spectroscopy (DRIFTS) experiments were frequently employed to detect the existence form of platinum species (single atoms, clusters or particles). As shown in Fig. 2d and Supplementary Fig. 6, only gases CO peaks was detected for 1Pt2Bi-SiO$_2$. However, for 1Pt-SiO$_2$-400 with almost identical oxide cluster size and loading amounts of platinum species (Supplementary Fig. 7) with 1Pt2Bi-SiO$_2$, two CO adsorption peaks on Pt$^{\delta+}$ single atom (2093 cm$^{-1}$), oxide cluster (2075 cm$^{-1}$)[19] were determined. It further evidences the formation of oxidized Pt$_x$Bi$_y$O$_z$ cluster, which possesses totally different structural composition and adsorption capacity with Pt$_x$O$_z$ cluster in Fig. 2e. To our best knowledge, it is the first time to observe the formation of uniform platinum-bismuth oxide clusters to suppress the aggregation of Pt species.

**Catalytic performance of Pt/PtBi–SiO$_2$ catalysts in CO oxidation.** CO oxidation was applied as a model reaction to investigate the role of bismuth-dopant. To mimic lean-burn diesel engine exhaust and acquire the best catalytic performance, we used excess O$_2$ in the reactant (CO/O$_2$ = 1/20)[30]. A gas hourly space velocity of 134,000 ml g$_{cat}$$^{-1}$ per hour was tried to match standard vehicle exhaust conditions. When the catalysts were pretreated at 300 °C under air, Bi-free and Bi-promoted samples show almost same CO oxidation activity with complete CO conversion at ~220 °C (Supplementary Fig. 8), may due to poor ability to adsorb CO or overhigh valence of platinum species[31]. However, we found that hydrogen reduction significantly enhanced CO oxidation activity for Bi-promoted catalysts (Fig. 3a). The temperature of 50% CO conversion dropped off from 165 to 85 °C as the reduction temperature increasing from 0 to 210 °C. Interestingly, a platform appeared as hydrogen reduction at 150 and 180 °C, indicating a structure transformation of active site occurred during the hydrogen reduction compared with fresh Bi-doped catalysts. On the basis of the CO oxidation activity (Fig. 3b), a remarkable promotion to platinum-silica catalysts was observed by the addition of Bi oxide species with similar Pt loading (0.9 wt. %). The catalytic performance reaches the maximum at the dopant of 2 wt.% (Supplementary Fig. 9), due to overmuch bismuth oxide species hindering CO adsorption or covering platinum active site. For comparison, pure Bi catalyst (2Bi–SiO$_2$) shows no CO oxidation activity below 160 °C (Fig. 3b), demonstrating bismuth species is not active site, just as secondary dopant to modification platinum active site. Furthermore, we collected the kinetic data to compare the inherent catalytic activity. The specific activity normalized by the platinum amount for 1Pt2Bi-SiO$_2$ was 487 µmol$_{CO}$ g$_{Pt}$$^{-1}$ s$^{-1}$ at 110 °C, as active as the reported Pt/CeO$_2$ catalysts (103−518 µmol$_{CO}$ g$_{Pt}$$^{-1}$ s$^{-1}$ at 80 −130 °C, see Table 1), as well as 10 times higher than that of pure

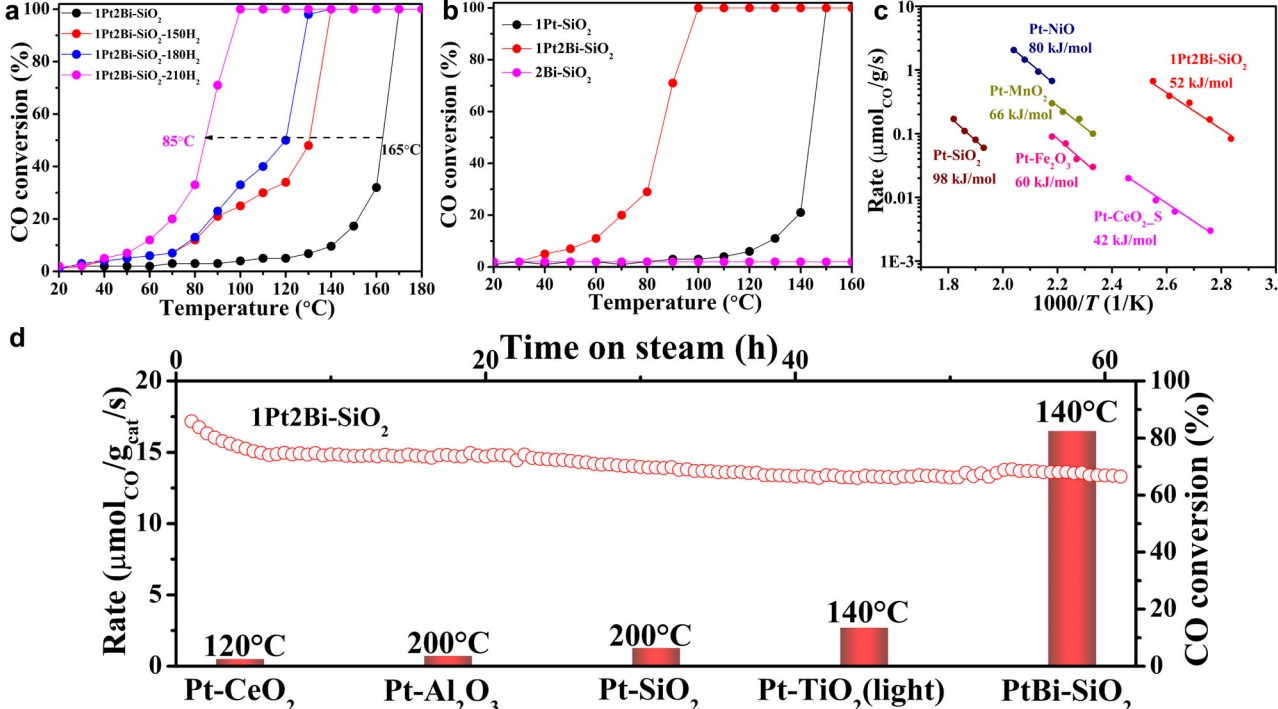

**Fig. 3 Catalytic performance of Pt/PtBi-SiO$_2$ catalysts in CO oxidation. a** CO oxidation experiments with different H$_2$-pretreatment temperature for 1Pt2Bi-SiO$_2$; **b** Catalytic CO oxidation light-off performance of different catalysts at a gas hourly space velocity of 134,000 mL g$_{cat}$$^{-1}$ h$^{-1}$ in 1 vol.% CO/20 vol.% O$_2$/79 vol.% N$_2$. Temperature ramp: 2 °C min$^{-1}$; **c** Apparent activation energy ($E_a$) of various catalysts; **d** CO oxidation stability test (150 °C, 200,000 mL g$^{-1}$ h$^{-1}$) and reaction rate of different catalysts.

**Table 1 Comparison of the activities over the representative platinum catalysts for the oxidation of carbon monoxide.**

| | Temp (°C) | Gas feed composition | $r_1$ | $r_2$ | Apparent activation energy (kJ/mol) | Ref. |
|---|---|---|---|---|---|---|
| *Inert support* | | | | | | |
| 1 wt.%Pt/θ-Al$_2$O$_3$ | 200 | 3.7%CO/3.7%O$_2$/He | 0.7 | 72 | – | 57 |
| 0.2 wt%Pt/m-Al$_2$O$_3$ | 200 | 2.5% CO/2.5% O$_2$/Ar | 0.2 | 118 | 78 | 20 |
| 0.5 wt.%Pt/SiO$_2$ | 200 | 40TorrCO/100 TorrO$_2$/He | 1.3 | 256 | 95 | 4 |
| *Reducible support* | | | | | | |
| 1 wt.% Pt/CeO$_2$ | 225 | 2%CO/1%O$_2$/N$_2$ | 8.7 | 871 | 55 | 32 |
| 1.1 wt%Pt/TiO$_2$ | 140 | 1%CO/10%O$_2$/N$_2$ | 2.9 | 261 | – | 58 |
| 0.5 wt.%Pt/MnO$_2$ | 200 | 40TorrCO/100Torr O$_2$/He | 5.5 | 1107 | 66 | 4 |
| 0.5 wt.%Pt/Fe$_2$O$_3$ | | | 1.7 | 349 | 70 | |
| 0.5 wt.%Pt/CeO$_2$ | | | 6.9 | 1384 | 63 | |
| 1 wt%Pt/CeO$_2$ | 134 | 0.4%CO/10%O$_2$/He | 1.0 | 103 | 43 | 2 |
| 1 wt%Pt/CeO$_2$ | 80 | 1.9%CO/1.2%O$_2$/He | 5.2 | 518 | 30 | 59 |
| 0.22Pt/CeO$_2$ | 110 | 1%CO/20%O$_2$/He | 0.2 | 92 | – | 23 |
| Our results | | | | | | |
| 1Pt2Bi-SiO$_2$ | 90 | 1%CO/20%O$_2$/N$_2$ | 1.7 | 210 | 50 | This work |
| | 100 | | 3.1 | 390 | | |
| | 110 | | 3.9 | 487 | | |
| | 175 | 2%CO/1%O$_2$/N$_2$ | 9.1 | 1138 | 47 | |
| | 200 | | 15.6 | 1953 | | |
| | 225 | | 58.3 | 7289 | | |

$r_1$ reaction rate by catalyst weight ($\mu mol_{CO}\,g_{cat}^{-1}\,s^{-1}$), $r_2$ reaction rate by Pt amount ($\mu mol_{CO}\,g_{Pt}^{-1}\,s^{-1}$).

Pt catalyst (Supplementary Table 4). For eliminating size-effect on active site, 1Pt–SiO$_2$-400 was similarly inactive for CO oxidation reaction with complete conversion of CO at 200 °C (Supplementary Fig. 9b), though possessing similar cluster size (1.8 ± 0.1 nm) to 1Pt2Bi–SiO$_2$ (Supplementary Fig. 7). It indicates that the dispersion of platinum species does not dominantly determines the CO oxidation activity. In another hand, the apparent activation energy ($E_a$) of Bi-promoted platinum catalysts (~52 kJ mol$^{-1}$) is similar to that of CeO$_2$-supported Pt catalyst (40−50 kJ mol$^{-1}$)[2,32], and distinctly lower than that of pure 1Pt-SiO$_2$ (~70 kJ mol$^{-1}$) and other inert support platinum catalysts[5,33] (Fig. 3c). This may give a hint on the totally different reaction mechanism or active site for our Bi-promoted Pt-SiO$_2$ catalysts. Additionally, 1Pt2Bi-SiO$_2$ showed remarkable thermo-stability for ~ 70 h at 150 °C and 200,000 mL g$^{-1}$ h$^{-1}$ in 1%CO/ 20%O$_2$/N$_2$ (Fig. 3d) and catalytic stability at high temperature (500 °C) under the high space velocity to maintain the CO conversion at 95% (300,000 mL g$_{cat}^{-1}$ h$^{-1}$, Supplementary Fig. 9c).

**Structural characterization of used Pt/PtBi-SiO$_2$ catalysts.** After CO oxidation, we employed a comprehensive characterization to determine the actual structure of active site. The aberration-corrected HAADF-STEM images (Fig. 4a–c and Supplementary Fig. 10) showed that the platinum species was still in the formation of cluster (~2 nm) without any visible lattice fringes of crystal Pt/PtO/PtO$_2$ component after CO oxidation. Meanwhile, the related STEM-EDS mapping results of cluster indicated that the Pt and Bi elements are still distributed together, not core-shell at the same area (Fig. 4d and Supplementary Fig. 11). Thus, the bismuth element around platinum cluster still makes an interaction with platinum species to prevent aggregation of clusters into huge particles, even after hydrogen reduction. In another hand, there is an obvious aggregation of platinum cluster (~3.0 nm) compared to fresh 1Pt-SiO$_2$ (~1.7 nm) in Supplementary Fig. 10c. Mahmudov and co-workers found obvious aggregation of platinum particles on activated carbon after hydrogen reduction[34]. The XRD results in Fig. 4e also confirmed the huge metallic platinum particles were still maintained in 1Pt-SiO$_2$ with sharp diffraction peaks at 39.7° and 46.2°. In contrast,

no obvious diffraction peaks of Pt/PtO/PtO$_2$ phases were detected in Bi-promoted samples, further confirming the high dispersion of platinum species. There was a broad peak of Bi$_2$O$_3$ in 1Pt5Bi-SiO$_2$, due to the aggregation of bismuth species.

According to the catalytic performance in CO oxidation with hydrogen reduction pretreatment, an obvious structural evolution occurred in Bi-promoted catalysts. Recently, more and more reports indicate that in-situ or *quasi* in-situ techniques cloud seize the evolution of active site and the synergistic effect on bimetal catalysts[35–38]. The quasi in-situ XAFS experiment could acquire a good signal-noise ratio of XAFS spectrum for low content of metal and retain the real structure of active site under different condition. In order to elucidate the real active site structure in reductive and CO oxidation atmosphere, XAFS spectrums of 1Pt–SiO$_2$ and 1Pt2Bi–SiO$_2$ were collected after hydrogen reduction at different temperature (150 and 210 °C) and 1 h of time-on-stream in CO oxidation (100 °C) in a stainless reactor with two globe valves and tableting in glove box under nitrogen atmosphere and ambient temperature for further XAFS experiments without exposure to air.

Combination with the catalytic performance in Fig. 3a and previous reported[31,39], the lower oxidized state of Pt species is appropriate for lower temperature CO oxidation. XANES data in Fig. 5a, c indicated that the valance state of platinum species is decreasing (+1.2 to +0.2 and +2.5 to +0.4 for 1Pt-SiO$_2$ and 1Pt2Bi-SiO$_2$ respective) as the increasing of hydrogen reduction temperature (150–210 °C) (Supplementary Table 5). The corresponding XANES profiles during CO oxidation following the hydrogen reduction at 210 °C were collected in Fig. 5a, c, the average valance of platinum slightly increases from +0.2 to +1.0 and +0.4 to +1.3 for 1Pt–SiO$_2$ and 1Pt2Bi–SiO$_2$ respective compared with that in reduced state at 210 °C. It demonstrated that the oxygen-rich reaction gas could make platinum species slight oxidative. Meanwhile, the similar valance of platinum species about +1 indicates that the difference in CO oxidation activity for 1Pt-SiO$_2$ and 1Pt2Bi-SiO$_2$ is not mainly due to platinum valance. Furthermore, the relevant EXAFS profiles were exhibited in Fig. 5c, d. For 1Pt–SiO$_2$, after hydrogen reduction, a main metallic Pt−Pt shell ($R \approx 2.75$ Å, $CN \approx 7.0$-9.2) was acquired and only a minor Pt−O shell ($R \approx 2.00$ Å, $CN \approx 0.6$) can be fitted

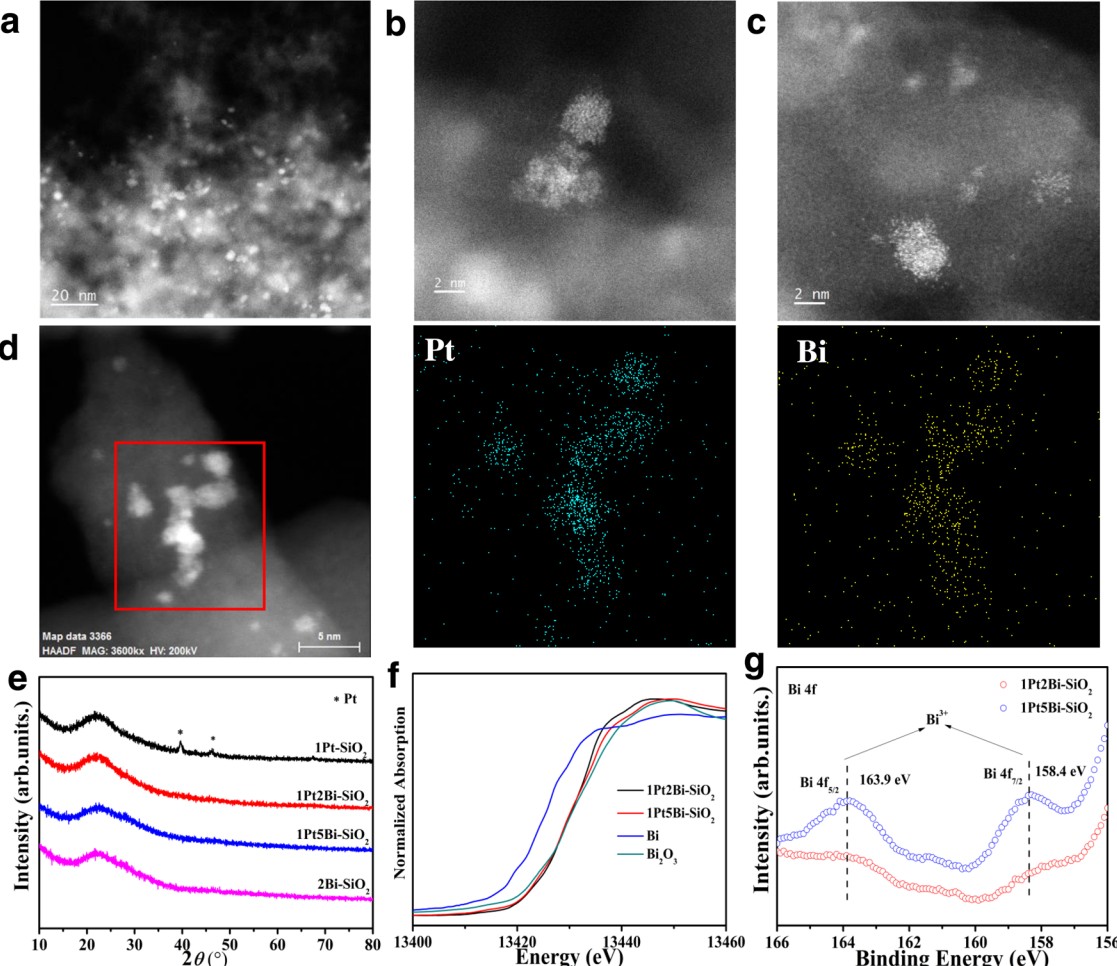

**Fig. 4 Structural characterization of used Pt/PtBi-SiO$_2$ catalysts.** Representative aberration-corrected HAADF-STEM images of **a–c** for used 1Pt2Bi-SiO$_2$ at different scales and **d** the corresponding EDS element mapping of used 1Pt2Bi-SiO$_2$; **e** XRD patterns; Bi L$_3$–edge **f** XANES profiles and **g** *XPS* profiles of used Pt/PtBi–SiO$_2$ samples.

at low reduction temperature (150 °C), which may result in low activity due to no surface-active oxygen to participate in CO oxidation[40]. For 1Pt2Bi-SiO$_2$, in order to require more reliable local coordination structure, we conducted the EXAFS fitting process with or without Pt−O−Bi shell in Supplementary Fig. 13. Obviously, Pt−O−Bi shell is an essential composition to acquire the most reasonable fitting results. In Fig. 5d and Supplementary Table 5, there is an obvious difference in local coordination structure compared with 1Pt-SiO$_2$. When the reduction temperature is at 150 °C, only partial platinum species is reduced to metallic Pt with a small $CN \approx 4.8$ at $R \approx 2.75$ Å, plus strong Pt−O ($R \approx 2.00$ Å, $CN \approx 2.1$) and Pt−O−Bi ($R \approx 2.98$ Å, $CN \approx 1.8$) shells. It proves the CO conversion platform of 1Pt2Bi-SiO$_2$ in Fig. 3a is due to incomplete evolution of active site at 150 °C. Furthermore, there is similar coordination structure for 1Pt2Bi-SiO$_2$-210 °C and 1Pt2Bi–SiO$_2$–CO oxidation, indicating the reduction temperature at 210 °C is appropriate for construction of optimal active sites. The EXAFS fitting results for Pt−Pt shell with $CN \approx 7.4$ also confirmed the average grain size of platinum cluster was ~ 2 nm for 1Pt2Bi–SiO$_2$[41] as observed in HAADF-STEM images (Fig. 4a–c). In addition, we found that the Pt−O ($R \approx 2.00$ Å, $CN \approx 1.5$) of 1Pt2Bi-SiO$_2$-CO oxidation is higher than that ($R \approx 1.98$ Å, $CN \approx 0.6$) of 1Pt2Bi-SiO$_2$-210 °C, may due to the formation of more Pt−[O]$_x$−Bi structure in oxidative atmosphere. And the detection of Pt−O−Bi ($R \approx 3.00$ Å, $CN \approx 2.3$) composition (Fig. 5d and Supplementary Table 3)

also validates the existence of Pt−[O]$_x$−Bi structure. Thus, the clusters in 1Pt2Bi–SiO$_2$ observed in aberration-corrected HAADF-STEM images (Fig. 4a–c) is metallic platinum cluster. In another hand, the Bi L$_3$ XANES and Bi 4f *XPS* profiles in Fig. 4f, g indicate that bismuth species is still in oxidative state even after hydrogen reduction and CO oxidation, which can exclude the formation of PtBi alloy. As a reference, 1Pt–SiO$_2$-400 exhibited low activity in CO oxidation (Supplementary Fig. 9b), even though possessing similar local coordination structure for Pt−O and Pt−Pt shell to the used 1Pt2Bi–SiO$_2$ (Supplementary Fig. 14b). Thus, taking the consideration of the similar valance state for Pt in 1Pt–SiO$_2$ and 1Pt2Bi-SiO$_2$ (~+1.0), we can draw a conclude that the surface Pt−[O]$_x$−Bi structure plays a key role in low temperature CO oxidation reaction rather than solo oxidized Pt$_x$O$_z$ cluster.

**The reducibility and active oxygen for Pt/PtBi-SiO$_2$.** As we know, the reducibility of catalysts is crucial in various redox reactions[42–44]. For fresh samples, a main reduction peak located at ~ 100 °C appeared on profiles of H$_2$−temperature programmed reduction (H$_2$−TPR) in Fig. 6a for 1Pt−SiO$_2$ contributed by the reduction of Pt$_x$O$_z$ clusters[45]. However, for Bi-doped samples, the first broad reduction peak was shifted to 162 °C (1Pt2Bi–SiO$_2$) and 197 °C (1Pt5Bi–SiO$_2$) in Supplementary Fig. 15a, due to the strong interaction of Pt−O−Bi[46], as confirmed by EXAFS fitting

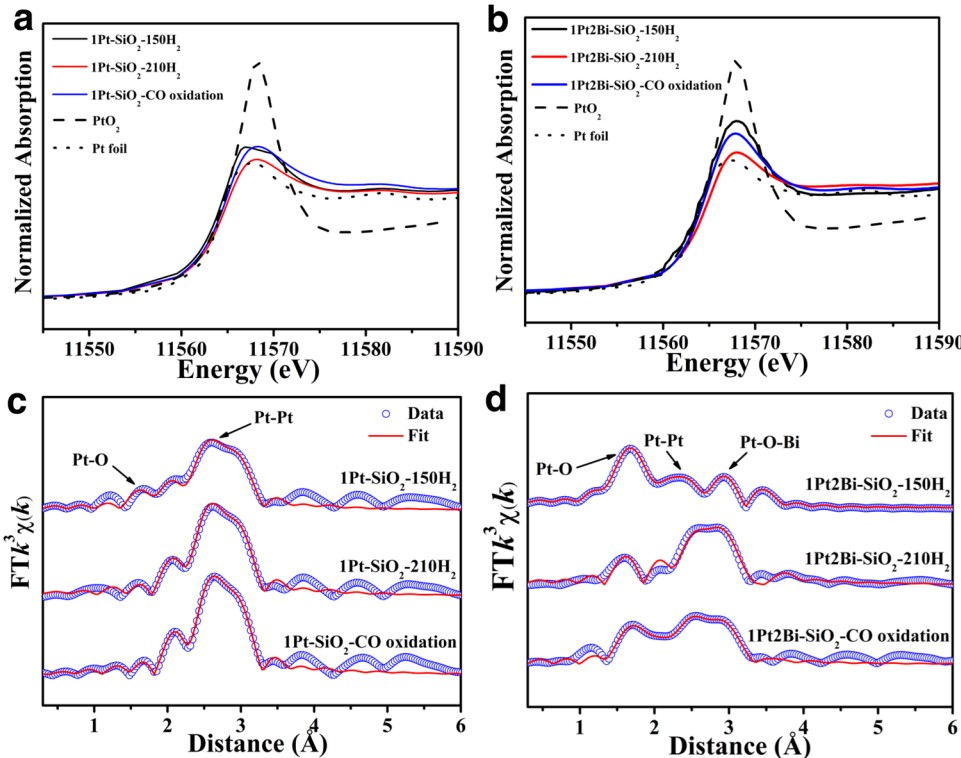

**Fig. 5 The local coordination structure of Pt/PtBi–SiO$_2$ catalysts.** *Quasi* in-situ Pt L$_3$–edge **a**, **c** XANES profiles and **b**, **d** EXAFS fitting results in *R* space for 1Pt-SiO$_2$ (**a**, **b**) and 1Pt2Bi-SiO$_2$ (**c**, **d**) under specific conditions.

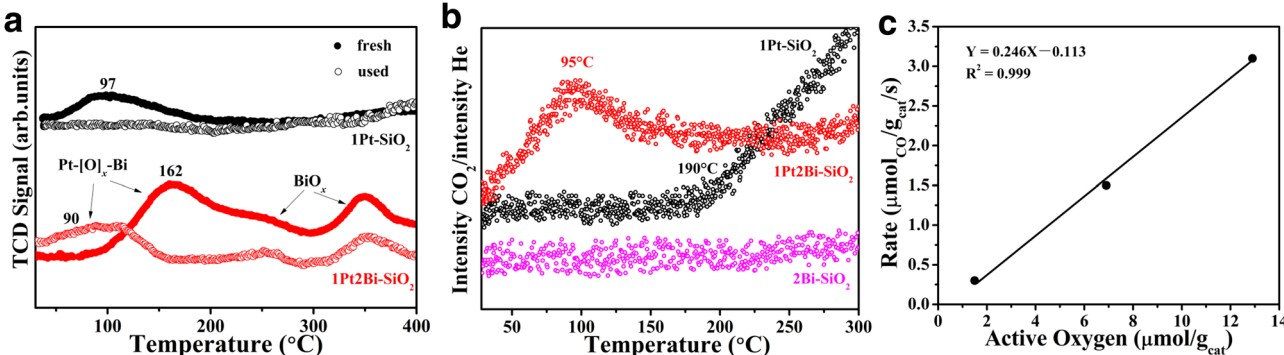

**Fig. 6 The reducible property and active oxygen for Pt/PtBi–SiO$_2$.** **a** H$_2$−TPR profiles of fresh Pt/PtBi–SiO$_2$ catalysts and used catalysts after CO oxidation without exposure to air; **b** in-situ CO−TPR of used Pt/PtBi–SiO$_2$ samples after CO oxidation without exposure to air; **c** linear equation for active oxygen vs reaction rate for used Pt/PtBi–SiO$_2$ samples.

results. The hydrogen consumption values (Supplementary Table 6) increased for 107 µmol/g (1Pt–SiO$_2$) to 185 µmol/g (1Pt2Bi–SiO$_2$). Although the doping of bismuth oxide species can enhance the surface oxygen, there is no promotion in CO oxidation. Because the over-strong interaction between platinum and bismuth (Pt−O−Bi) cannot release oxygen atom to take part in reaction. CO−temperature programmed reduction (CO−TPR) results in Supplementary Fig. 16 also demonstrated the surface-active oxygen for 1Pt2Bi-SiO$_2$ (~75 µmol/g) is almost two times than that (~40 µmol/g) of 1Pt–SiO$_2$. However, these oxygen species only reacted with CO molecule above 100 °C, well consistent with low activity in CO oxidation with oxidative pretreatment (Supplementary Fig. 8).

We also carried out the H$_2$−TPR and CO−TPR experiments for used Pt/PtBi–SiO$_2$ samples after CO oxidation without air exposure absolutely to detect reducibility and active oxygen of active site. For H$_2$−TPR experiments, no reduction peak below

200 °C appeared in 1Pt-SiO$_2$ due to hydrogen reduction treatment before reaction. However, a broad reduction peak starting from 50 °C still existed in Bi-doped samples, indicating the existence of Pt−[O]$_x$−Bi structure with low oxidation state of platinum. It indicates the unique active site of platinum cluster with surface Pt−[O]$_x$−Bi structure exhibits more reducible property to release oxygen. In addition, there are two reduction peaks at 250 and 350 °C, which are attributed to the high dispersion BiO$_x$ cluster adjoining to platinum cluster and isolated BiO$_x$ cluster deposited on the surface of SiO$_2$. However, the oxygen provided by these BiO$_x$ clusters only can be activated at high temperature (>200 °C), which makes few contributions to low temperature CO oxidation activity. Additionally, CO−TPR results of used 1Pt2Bi-SiO$_2$ also evidenced these surface oxygen atoms in Pt−[O]$_x$−Bi structure was superior active to react with CO molecule generating CO$_2$ from ~50 °C (Fig. 6b), well consistent with the CO oxidation "light off" temperature (Fig. 3b).

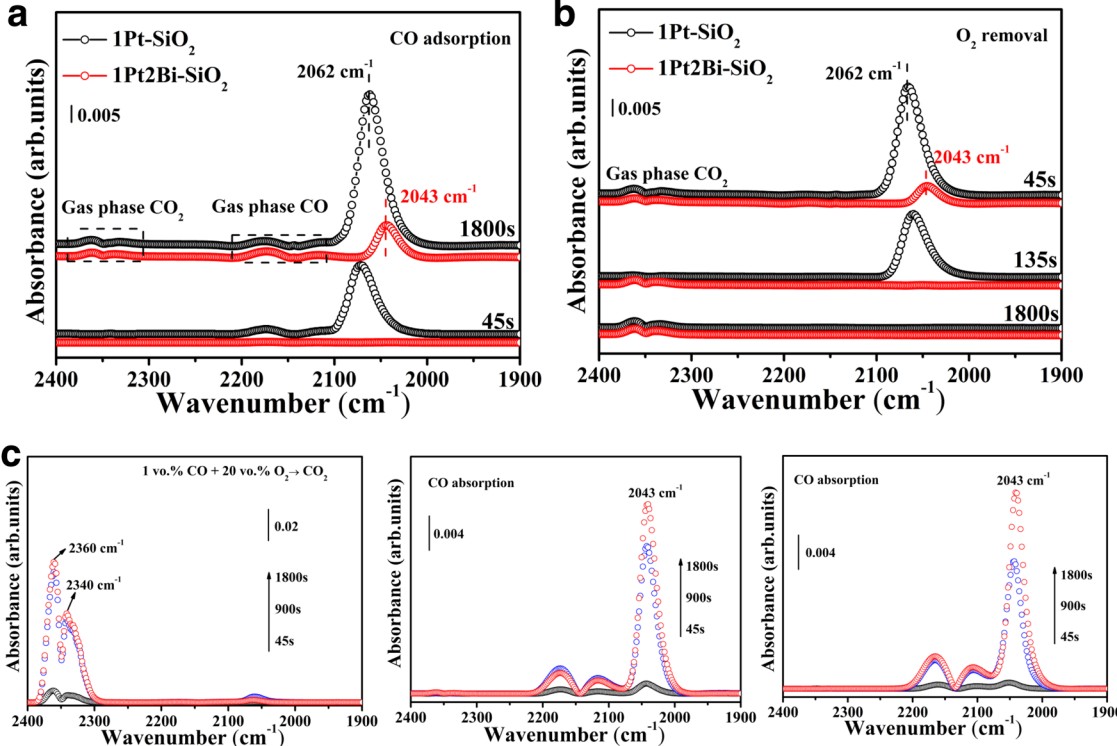

**Fig. 7 The CO adsorption on Pt−[O]$_x$−Bi active site.** In-situ DRIFTS study of **a** CO adsorption and **b** O$_2$ removal on Pt/PtBi-SiO$_2$ samples; **c** In-situ DRIFTS in the mode: "CO adsorption → reaction conditions (1% CO/ 20% O$_2$/N$_2$ flow) at 200 °C → CO adsorption" on 1Pt2Bi-SiO$_2$ without catalyst replacement and air exposure. (The catalysts were pretreated in-situ at 210 °C under H$_2$ flow in the DRIFTS reaction cell before data collection; CO flow rate: 30 mL min$^{-1}$; catalyst mass: 20 mg; temperature: 100 °C).

However, 1Pt–SiO$_2$ did not activate the surface hydroxyls to produce CO$_2$ (water-gas shift reaction) until 195 °C[11]. This highly reducible oxygen species may motivate initial CO oxidation (~50 °C) through Mars-van Krevelen (MvK) mechanism[47] and show a strong correlation between reaction rate and active oxygen amount in Fig. 6c. Therefore, the Pt−[O]$_x$−Bi structure with low valance state of Pt$^{\delta+}$ ($0 < \delta < 2$) species in the used Bi-promoted samples provides superior active oxygen species to catalyze preliminary stage of CO oxidation.

**The CO adsorption on Pt−[O]$_x$−Bi active site.** Despite the identification of active site structure in silica-supported platinum-bismuth catalysts being important, the adsorption of reaction gas is a more key factor for catalytic behavior. Because, recent reports have indicated that a lot of atomically dispersed platinum catalysts show low catalytic activity in CO oxidation due to over strong adsorption of CO, even with maximized number density of reaction sites[27,28,32,48]. Therefore, it is significant to investigate the CO adsorption on active site for Bi-free and Bi-promoted catalysts. The in-situ DRIFTS experiments displayed that CO adsorption intensity of Bi-promoted sample is moderate compared to that of pure platinum sample (Fig. 7a), indicating the surface Pt−[O]$_x$−Bi structure could prevent oversaturated adsorption of CO molecule (CO poison) on Pt clusters or nanoparticles, just like the reported alkali-doped Pt catalysts[11]. The CO adsorption reached saturation rapidly for 1Pt–SiO$_2$ at 2062 cm$^{-1}$ attributed to linear CO adsorbed on Pt$^0$ sites (Pt$^0$−CO)[49] (Fig. 7a and Supplementary Fig. 17), resulting in no other Pt sites dissociating gases oxygen into active oxygen species[19]. However, the peak for CO adsorption over 1Pt2Bi-SiO$_2$ was detected at 2043 cm$^{-1}$, which was absolutely different linear CO adsorbed on Pt$^0$ sites. This red-shift phenomenon can be eliminated the possibility of either size-effect of Pt species or CO

adsorption on pure Bi species. In one hand, 1Pt–SiO$_2$-400 with total platinum cluster also showed the peak ~2060 cm$^{-1}$ (Supplementary Fig. 18) similar with the peak (2062 cm$^{-1}$) in 1Pt–SiO$_2$. In another hand, there is no CO adsorption peak for 2Bi–SiO$_2$ except for gases CO peaks (Supplementary Fig. 19), no matter with oxidative or reduced pretreatment. Therefore, the low-frequency band at 2043 cm$^{-1}$ is attributed to CO molecules adsorbed on the unique active site (Pt−[O]$_x$−Bi), even less doping of bismuth species (Supplementary Fig. 20), and this may result from the unique local coordination structure of Pt−[O]$_x$−Bi structure or electron transfer from platinum atom to CO molecule[50,51]. Panagiotopoulou and co-workers reported that the alkali additives to Pt/TiO$_2$ catalyst generate a low-frequency shoulder band (~2030 cm$^{-1}$) resulting in strengthen of Pt−CO bond[51]. Furthermore, when the oxygen was introduced, the activated CO molecule adsorbed on Pt−[O]$_x$−Bi structure could be converted to CO$_2$ quickly (~135 s) and completely (Fig. 7b). Our density functional theory (DFT) calculation builds up various simulation model for Pt@PtO$_x$ without and with Bi dopant (Supplementary Fig. 21) and the corresponding calculated vibrational frequencies of a CO molecule adsorbed on different sites. The vibration peaks at 2062 cm$^{-1}$ and 2043 cm$^{-1}$ in the experiments were attributed to the configurations in Supplementary Figs. 21c, f respectively. One can see that upon the doping of the Bi atoms, the number of the Pt–O bonds had decreased. It is not surprising since Bi atoms are oxophilic and can seize O atoms from Pt. In Supplementary Fig. 22, we presented the density of states (DOS) for the $d$ electrons of the Pt atom (Supplementary Fig. 21c, f) on which the CO molecule adsorbs. As the coordination number of the Pt atom decreases, the center of the $d$ band becomes closer to the Fermi level, corresponding to strengthened activity. Thus, due to the unique structure of Pt−[O]$_x$−Bi, more electron was transformed from Pt

atom to CO molecules to activated carbon monoxide, resulting in the red-shift on CO adsorption band in DRIFTS experiments.

Additionally, we employed in-situ DRIFTS experiments with a mode: "CO adsorption → reaction conditions (1 vol.% CO/20 vol.% $O_2/N_2$ flow) at 200 °C → CO adsorption" to detect the stability of active site. For 1Pt2Bi-SiO$_2$, the Pt−[O]$_x$−Bi structure could be maintained after CO oxidation (2042 cm$^{-1}$), well consistent with the band after hydrogen pretreatment at 210 °C (Fig. 7c). For 1Pt-SiO$_2$, after CO oxidation, the CO adsorption band occurred a blue shift (2072 cm$^{-1}$) compared to the band (2062 cm$^{-1}$) after hydrogen reduction at 210 °C (Supplementary Fig. 23), due to the surface oxidation of metallic Pt cluster.

## Discussion

In summary, we have prepared silica-supported platinum-bismuth catalysts via an incipient wetness impregnation, possessing excellent sinter resistance due to the formation of oxidized Pt$_x$Bi$_y$O$_z$ cluster. The Bi-promoted catalysts exhibit an absolutely different active site for platinum cluster with surface Pt−[O]$_x$−Bi structure compared with pure platinum sample, providing superior active oxygen species activated by CO at low temperature (~50 °C) with a high $CO_2$ production rate of 487 $\mu mol_{CO2}$ $g_{Pt}^{-1}$ s$^{-1}$ at 110 °C. Even after hydrogen reduction, the surface Pt−[O]$_x$−Bi structure still stabilizes platinum cluster in cluster-scale with unique properties: (1) preventing oversaturated CO adsorption from poison of platinum species; (2) activating CO molecules to catalyze CO oxidation in a lower apparent activation energy. Therefore, we have provided a general approach towards design of potential active and stable platinum catalysts.

## Methods

**Catalyst preparation**. SiO$_2$ was calcination at 400 °C for 4 h (ramping rate: 2 °C/min) to remove surface water before catalysts preparation. Deposition of platinum, and bismuth onto the SiO$_2$ support was carried out by a co-incipient wetness impregnation. Firstly, a solution of Bi(NO$_3$)$_3$ (50−124 mg respective) and Pt (NH$_3$)$_4$(NO$_3$)$_2$ (20 mg) in 0.1 mol/L HNO$_3$ solutions (3 mL) was added dropwise onto SiO$_2$ power (1 g) under manually stirring. The powders were standing in ambient conditions for 2 h and then dried in still air at 80 °C for 12 h, followed by air-calcination at 550 °C for 4 h (ramping rate: 2 °C/min). As comparison, we also prepared 1Pt–SiO$_2$ calcinated at 400 °C, 1Pt1Bi–SiO$_2$, and 2Bi–SiO$_2$ samples with same synthetic method. The Bi and Pt contents were controlled on demand during preparation process of catalysts, and the data of these catalysts is as follows:

1Pt–SiO$_2$ (1 wt. % Pt, 0 wt. % Bi),
1Pt2Bi–SiO$_2$ (1 wt. % Pt, 2 wt. % Bi),
1Pt5Bi–SiO$_2$ (1 wt. % Pt, 5 wt. % Bi)
2Bi–SiO$_2$ (2 wt. % Bi)
1Pt–SiO$_2$-400 (1 wt. % Pt)
1Pt1Bi–SiO$_2$ (1 wt. % Pt, 1 wt. % Bi).

**Catalytic activity tests**. The CO oxidation activities for the PtBi–SiO$_2$ samples were evaluated in a plug flow reactor using 30 mg of sieved (20−40 mesh) powders in a gas mixture of 1 vol.% CO/20 vol.% O$_2$/N$_2$ (from Jinan Deyang Corporation, 99.997% purity) at a flow rate of 67 mL/min giving a gas hourly space velocity (GHSV) of ~134,000 mL g$_{cat}^{-1}$ h$^{-1}$. The catalysts were pretreated in 5% H$_2$/N$_2$ (50 mL/min) at 210 °C for 30 min before reaction with 10 °C min$^{-1}$. After the catalysts cooled down to room temperature under a flow of pure N$_2$ gas, reactant gases were passed through the reactor. The outlet gas compositions of CO and CO$_2$ were monitored online by nondispersive IR spectroscopy (Gasboard 3500, Wuhan Sifang Company, China). CO conversion was defined as CO$_{reaction}$/CO$_{input}$ × 100%. The related stability tests were done in the same conditions at the constant reaction temperature of 100 °C for 10 h with a GHSV of ~134,000 mL g$_{cat}^{-1}$ h$^{-1}$. Rate measurements were made in the separate catalytic tests rather than the "light-off" mode, i.e., the same gas composition, but at specific space velocities to ensure operation in the kinetic regime (<20% conversion of CO).

**Materials characterization**. The actual platinum and bismuth concentrations of the catalysts were analyzed by inductively coupled plasma atomic emission spectroscopy (ICP-AES; Optima 5300DV, PerkinElmer). The air-calcined samples (fresh catalysts) were used directly for characterization. First, 0.1 g catalyst (accurate to 0.0001 g) was added to 2 mL hydrofluoric acid under continuous stirring until the powder was dissolved adequately. Second, the as-formed SiF$_4$ was removed via evaporation. Then, almost 3 mL of nitric acid was introduced and the

solution was kept slightly boiling for 2 h. Finally, the solution was cooled to nearly 25 °C and diluted for the ICP-AES test.

XRD patterns were recorded on a Bruker D8 Advance diffractometer (40 kV, 40 mA) with a scanning rate of 4° min$^{-1}$, using Cu $K_{\alpha 1}$ radiation ($\lambda = 1.5406$ Å). The diffraction patterns were collected from 10 to 80° with a step of 0.02°. The 2$\theta$ angles were calibrated with a μm-scale Alumina disc. The powder sample after grinding was placed inside a quartz sample holder for each test. XPS analysis was performed on an Axis Ultra XPS spectrometer (Kratos, U.K.) with 225 W of Al $K\alpha$ radiation. The C 1 s line at 284.8 eV was used to calibrate the binding energies.

The nitrogen adsorption-desorption measurements were performed on an ASAP 2020-HD88 analyzer (Micromeritics Co. Ltd.) at 77 K. The measured powders were degassed at 250 °C under vacuum (< 100 μmHg) for over 4 h. The BET specific surface areas ($S_{BET}$) were calculated from data in the relative pressure range between 0.05 and 0.20. The pore diameter ($D_p$) distribution was calculated from the adsorption branch of the isotherms, based on the BJH method.

The TEM and high-resolution TEM (HRTEM) experiments were carried out on a FEI Tecnai G$^2$ F20 microscope operating at 200 kV. All the tested samples were suspended in ethanol, and then a drop of this dispersed suspension was placed on an ultra-thin (3−5 nm in thickness) carbon film-coated Cu grid. The as-formed sample grid was dried naturally under ambient conditions before loaded into the TEM sample holder. The aberration-corrected HAADF-STEM images were carried out on a JEOL ARM200F microscope equipped with probe-forming spherical-aberration corrector.

**XAFS experiments**. Pt L$_3$ ($E_0 = 11564.0$ eV) and Bi L$_3$ ($E_0 = 13419.0$ eV) edge was performed at BL14W1 beamline of Shanghai Synchrotron Radiation Facility (SSRF) operated at 3.5 GeV under "top-up" mode with a constant current of 260 mA. The XAFS data were recorded under fluorescence mode with a Lytle detector for Pt/PtBi-SiO$_2$ samples. Only X-ray absorption near-edge structure (XANES) data of Bi L$_3$ edge was collected due to the similar edge energies of Bi and Pt L$_3$ edge. The energy was calibrated accordingly to the absorption edge of pure Pt and Bi foil. Athena and Artemis codes were used to extract the data and fit the profiles. For the XANES part, the experimental absorption coefficients as function of energies $\mu(E)$ were processed by background subtraction and normalization procedures. Based on the normalized XANES profiles, the molar fraction of Pt$^{4+}$/Pt$^0$ and Bi$^{3+}$/Bi$^0$ can be determined by the linear combination fit with the help of various references (Pt foil for Pt$^0$, PtO$_2$ for Pt$^{4+}$, Bi foil for Bi$^0$ and Bi$_2$O$_3$ for Bi$^{3+}$). For the extended X-ray absorption fine structure (EXAFS) part, the Fourier transformed (FT) data in $R$ space were analyzed by applying PtO$_2$ and metallic Pt model for Pt−O and Pt−Pt contributions. The passive electron factors, $S_0^2$, were determined by fitting the experimental data on Pt and Bi foils, and fixing the coordination number (CN) of Pt−Pt and Bi−Bi to be 12, and then fixed for further analysis of the measured samples. The parameters describing the electronic properties (e.g., correction to the photoelectron energy origin, $E_0$) and local structure environment including coordination number (CN), bond distance (R) and Debye-Waller factor around the absorbing atoms were allowed to vary during the fit process. To distinguish the effect of Debye-Waller factor from coordination number, we set $\sigma^2$ to be 0.0030 and 0.0080 Å$^2$ for all the analyzed Pt−O and Pt−Pt shells, according to the fitted results of Pt foil and PtO$_2$ standards. We also set $\sigma^2$ to be 0.008 Å$^2$ for all the analyzed Pt−Bi paths, considering the Z number of Bi (83) is close to that of Pt (78). To distinguish the effect of correction to the photoelectron energy origin from distance, we set $\Delta E_0$ to be 11.3, 13.3, and 14.1 eV for fresh 1Pt-SiO$_2$, 1Pt2Bi-SiO$_2$, and 1Pt5Bi-SiO$_2$, respectively, which were obtained from the linear combination fits on XANES profiles and the fitting results of Pt foil ($\Delta E_0 = 8.3 \pm 1.2$ eV) and bulk PtO$_2$ ($\Delta E_0 = 15.0 \pm 0.9$ eV) standards and the $\Delta E_0$ for all used samples is 7.4 $\pm$ 1.8 eV according to EXAFS fitting results. For quasi in-situ XAFS experiment, the samples were pretreated in a stainless reactor with two globe valves and transferred to a glove box under nitrogen atmosphere and ambient temperature for tabletting without exposure to air.

**In-situ DRIFTS**. It was carried out in a diffuse reflectance cell (Harrick system) equipped with CaF$_2$ windows on a Bruker Vertex 70 spectrometer using a mercury-cadmium-telluride (MCT) detector cooled with liquid nitrogen. In a typical steady test, the powder sample (ca. 20 mg) was pretreated in air (21 vol% O$_2$/79 vol% N$_2$) at 300 °C or in hydrogen gas (5 vol.% H$_2$/He) at 210 °C for 30 min and cooled to room temperature under pretreatment atmosphere (30 mL min$^{-1}$). Then a background spectrum was collected via 32 scans at 4 cm$^{-1}$ resolution. The reaction gas with 1% CO/20% O$_2$/79% N$_2$ was introduced into the in-situ chamber (30 mL min$^{-1}$) and heated in a stepped way (every 40 K); DRIFTS spectra were obtained by subtracting the background spectrum from subsequent spectra. The IR spectra for every step were recorded continuously for 40 min to reach the equilibrium. Analysis of the spectra has been carried out by using OPUS software. For further investigation of the process of adsorption-desorption of CO over Pt/PtBi–SiO$_2$ catalysts, a "CO − N$_2$−CO−O$_2$" test was measured with in situ DRIFTS. The process of activation was carried out as described above. Then a background spectrum was collected at a certain temperature (100 °C) under pure N$_2$ (30 mL min$^{-1}$). The catalyst was exposed continuously to 2% CO in N$_2$ for CO adsorption for 30 min. Once CO gas was switched to an N$_2$ stream, also the corresponding IR spectra were recorded for 30 min. Then the catalyst was exposed to 2% CO in N$_2$ for CO re-adsorption for 30 min; ultimately 1% O$_2$ in an N$_2$ stream was introduced,

in order to follow the surface changes during the CO removal process. The CO adsorption for Pt/PtBi-SiO$_2$ samples was also carried out in the mode "CO adsorption → CO oxidation at 200 °C → CO adsorption" to detect structure evolution of adsorption site after CO oxidation.

**DFT calculations**. First-principles calculations were performed within the framework of the density functional theory (DFT) using the Vienna ab initio simulation package (VASP)[52,53]. The projector augmented wave (PAW) method was employed[54], and the wave functions were expanded by plane-wave basis sets with an energy cutoff of 400 eV. The exchange-correlation effects were described by the optPBE-vdW functional[55,56], which explicitly takes Van der Waals interactions into account. The simulation model of the PtO$_x$ system was constructed by nesting an icosahedral Pt$_{55}$ cluster inside a platinum oxide layer that contains 32 Pt atoms and 66 O atoms, and the deposited Bi atoms (10 atoms in the model) were placed on the outer layer of the platinum oxide. The unit cell contains enough vacuum along all three directions to eliminate spurious interactions between periodic images. Since the model of the system is an isolated cluster, the first Brillouin zone was sampled using the Γ point only. In the geometry optimizations, all atoms were allowed to relax until the maximum force below 0.03 Å/eV. The vibrational frequencies of CO were calculated through a finite differential approach, in which only the CO adsorbate, the Pt atom that connects it, and the surrounding Pt, O and Bi atoms were allowed to move while all other atoms were kept frozen.

**Reducible property and surface oxygen**. Hydrogen temperature-programmed reduction (H$_2$–TPR) was applied to determine the pretreatment temperature under hydrogen and reducibility. The measurements were carried out on a Micromeritics Autochem II 2920 instrument. Fresh (as calcinated in air) catalysts were used for characterization. Prior to the measurement, the catalyst (20–40 mesh, ~100 mg) was pretreated for 30 min in a flow of 5% O$_2$/He (30 mL min$^{-1}$) at 300 °C (10 °C min$^{-1}$) for 30 min. The test was carried out from room temperature to 600 °C (10 °C min$^{-1}$) at a ramp of 10 °C min$^{-1}$ under 5% H$_2$/He (30 mL min$^{-1}$). A thermal conductivity detector (TCD) was used to detect the changes of hydrogen concentration. The in situ H$_2$-TPR for used samples was also performed on Autochem II 2920 instrument after CO oxidation without air exposure with He purging to remove reactant gas completely.

Carbon monoxide Temperature-programmed reduction (CO–TPR) experiments using CO as reductant were performed on Micromeritics Autochem II 2920 instrument with mass spectrometer (Tilon GRP Technology Limited LC-D200M) to detect the active oxygen species in fresh and used samples. The fresh sample (100 mg, 20–40 mesh) was pretreated in 5% O$_2$/He (30 mL min$^{-1}$) at 300 °C (10 °C min$^{-1}$) for 30 min. The used samples after CO completely oxidation was conducted for CO–TPR without air exposure and then cooling down to 30 °C and switching to He (50 mL min$^{-1}$) for 10 min to remove reactant gas (CO and O$_2$ molecules). Finally, the sample was heated from room temperature to 500 °C with a step of 5 °C min$^{-1}$ under a mixture gas flow of 5% CO/He (20 mL min$^{-1}$). During the measurement, the signals of He ($m/z = 4$), CO$_2$ ($m/z = 44$), H$_2$O ($m/z = 18, 17$), CO ($m/z = 28, 16$), and H$_2$ ($m/z = 2$) were detected. The reported CO$_2$ and H$_2$ data were normalized by dividing by the corresponding He standard signal ($m/z = 4$).

## Data availability
The authors declare that all other data supporting the findings of this study are available within the article and Supplementary information files, and also are available from the corresponding author on reasonable request.

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

## Acknowledgements

This work was supported by "National Key Basic Research Program of China" (2017YFA0403402); "Project of the National Natural Science Foundation of China" (21773288, U1932119, 21771117 and 21805167), the Taishan Scholar Project of Shandong Province of China, the Young Scholars Program of Shandong University and the Outstanding Youth Scientist Foundation of Hunan Province (2020JJ2001). This work was also supported by shanghai large Scientific facilities center (Y92G021221). We thank Dr. Shuang-quan Hu (Evonik Industries) for his kind help on supply of high surface-area silica support.

## Author contributions

B.N. developed platinum-bismuth catalysts, tested for catalytic activity, and prepared the article. M.S. carried out the TEM/HRTEM images for all catalysts. Q.F. and J.Y.L. made the DFT calculations. J.Y. finished partial characterization of catalysts. L.L.Z. and W.W.W. measured in-situ DRIFTS data. L.N.L. carried out XAS measurements and analyzed the data. C.M. carried out the aberration-correction HAADF-STEM and EDS mapping images of all catalysts. J.X.C. performed the CO−TPR experiments. R.S. and C.J.J. conceived and supervised the project, procured funds, and wrote the manuscript, with contributions from all authors.

## Competing interests

The authors declare no competing interests.
