## [Peer Review File · Nature Communications]

REVIEWER COMMENTS

Reviewer #1 (Remarks to the Author):

The paper of Nan et al. presents the synthesis of a novel highly active CO oxidation catalyst based on Pt_xBi_yO_z clusters. The paper is very comprehensive, rich of profound characterization of the complex atomic structure of the catalyst and convincing in the interpretation of the data.

In a similar report recently published (Meunier F.C. et al. *Angew. Chem. Int. Ed.* 2020, 59,2–9) was already discussed the positive effect of oxidized Pt species and clearly showed the reaction mechanism of CO oxidation over these sites. Even though the presented paper shows very interesting results it does not reach unfortunately the high novelty standards of this journal, which need to fit to a broad audience.

Therefore, I cannot recommend the publication on Nature Communications and I encourage the authors to submit their manuscript to a more specific journal (e.g. ACS Catalysis). Also, I suggest to improve the introductory part highlighting the major novelty of their work. So far, this does not come through.

Reviewer #2 (Remarks to the Author):

Manuscript number: NCOMMS-20-48606

In the manuscript by B. Nan et al., the experimental and theoretical results on the catalytic performances of platinum-bismuth cluster on silica under CO oxidation are presented. It was suggested that Pt surface with Pt–O–Bi structure is the active site for CO oxidation via providing moderate CO adsorption and activating CO molecules with electron transformation between the platinum atom and carbon monoxide. I find that this is quite an intriguing result that shows the synergy between Pt and Bi, which can be useful for the design of novel bimetal catalysts. I suggest that the paper be accepted after major revision. Below are my detailed comments and questions that need to be addressed.

1. Many figures are not clear. The text in the figures is not clearly visible (For example, Fig. 4d). Authors need to improve the quality of figures to improve the readability of data.
2. Recent studies suggest that the synergistic catalytic effect on bimetal catalysts is related to the formation of reactive metal-oxide interfacial structure based on several operando surface characterization [For example, C. H Wu et al. *Nature Catalysis* 2, 78 (2019); T. S. Kim et al. *ACS Catalysis* 10, 10459 (2020); H. Lee et al. *Nature Communications* 9, 2235 (2018)]]]. Operando surface characterization such as ambient pressure XPS or environmental TEM would be ideal for identifying the reaction intermediates on PtBi nanoclusters under the reaction conditions. This point and relevant works can be discussed.
3. My main question is regarding Pt–O–Bi structure. H₂–TPR data shown in Figure 5a shows the presence of BiO_x. In addition, XPS data shown in Figure 4i shows the moderate oxidation of Bi. Therefore, it can be assumed that Pt coexists with BiO_x clusters, and the metal-oxide interfaces could be

reactive species. DFT calculation shows that Pt–O–Bi structure is the active site for CO oxidation. However, I don't see clear evidence of the formation of Pt–O–Bi structure.

4. Figure 1 shows high-resolution TEM images and EDS mapping of elements of PtBi nanoclusters. I am wondering if PtBi is core-shell-like structure because of the surface segregation of one element. After the reaction, how does EDS elemental mapping change?

Reviewer #3 (Remarks to the Author):

The manuscript by Si and coworkers present the improved CO oxidation activity of Pt-Bi catalysts supported on SiO₂. The authors claim that their atomically designed Pt-O-Bi species facilitates CO oxidation, recording the high activity of 487 μmolCO₂·gPt⁻¹·s⁻¹ at 110 °C. Generating additional catalytic pathway in Pt particles supported on irreducible oxides is essential in designing active/efficient catalysts with inherently high structural robustness. It is well-known that the interfaces between Pt and reducible oxide supporting materials facilitate CO oxidation through the Mars-van Krevelen mechanism. On the other hand, the surface of Pt nanoparticles supported on irreducible oxide supports oxidizes CO through the Langmuir-Hinshelwood mechanism, which CO and O₂ should be coadsorbed on Pt and O₂ activation should be followed for subsequent oxidation of adjacent Pt-CO*, making this process quite sensitive to the environmental factors such as the CO/O₂ concentration ratio in the reaction feedstock and the size of Pt particles, which is correlated with their surface area/volume ratio.

This manuscript presents an interesting results on the promoting role of Bi dopants in two folds: stabilization of small Pt species and catalytic activation of the Pt surfaces. Although several aspects claimed by the authors should be supported by additional data or discussion, my first impression on this manuscript is quite positive, as the main message is clear and is supported by combinatorial spectroscopic analysis results, which are nicely integrated with other experimental findings. The revised version of the manuscript may qualify the publication criteria of Nat. Commun.

Comments:

1. Introduction should be revised to be more informative and persuasive. Overall, introduction of the current manuscript is not persuasive and lack of important information, making the audience difficult to understand the urgency and the scientific value of this work.

Several suggestions:

1-1) Deduce and present the motivation that drove the authors to design Pt-based catalysts supported on SiO₂ for CO oxidation by discussing more precisely the advantages/disadvantages of using irreducible oxides as a supporting material for Pt. It is well known that Pt nanoparticles supported on such irreducible oxides catalyze CO oxidation through the Langmuir-Hinshelwood mechanism, rather than the interface-mediated Mars-van Krevelen mechanism.

1-2) Present scientific reason of using Bi as a dopant.

1-3) In the 3rd paragraph, provide in-depth discussion on the relevant recent studies on the catalytic role of various interfaces. Refer to Nat. Catal. 2, 955-970 (2019) and ACS Catal. 8, 7368-7387 (2018) and references therein.

2. I see that large Pt particles were developed during calcination (also evidence by the XRD spectrum) in 1Pt-SiO₂. Presumably, the most Pt-like XANES profile of 1Pt-SiO₂ compared with the others containing Bi suggests that Pt particles were formed in 1Pt-SiO₂. Based on their EXAFS fitting profile result, the authors stated that platinum oxide clusters (line 108, on page 6) were formed together with large Pt particles in 1Pt-SiO₂. However, I note that the XAS spectra of small Pt species (clusters) or Pt single atoms, which make close and strong contact between Pt and the oxygen ion of the supporting oxide, could be largely shifted toward that of PtO₂. Therefore, it is cautious to state that Pt oxide clusters (rather than single atoms or small sub-nanometer sized Pt clusters) were formed on SiO₂. Moreover, it is highly likely that the Pt particles presented in Figure S2a were not physically separated from the sample before conducting further analyses. If this is the case, the authors may centrifuge the sample and collect the inherent XAS spectra of the small Pt species supported on SiO₂, which are not biased from Pt particles.

3. The authors showed that Bi phenomenologically prevents formation of Pt particles. I agree with this statement. However, additional discussion from thermodynamic or energetic perspectives (highly likely based on DFT calculation results) on how Bi disintegrates or stabilizes Pt should be provided.

4. The DRIFTS spectra of 1Pt-SiO₂ presented in Fig. 2 and Suppl. Fig. 6 are needed to be more carefully analyzed. The location of the CO-probe peak is dependent to the supporting material and pre- or post-treatment condition. I think the peak at 2075 cm⁻¹ is somewhat originated from metallic Pt species, and the peak centered at 2093 cm⁻¹ could be the one from the weakly (positively) charged ionic Pt of small Pt clusters, rather than one from Pt single atoms. Note that the ref. 23 reports the IR spectra of Pt supported on FeO_x. To the best of my knowledge, the CO-probe peaks of Pt single atoms supported on oxides usually appear at above 2100 cm⁻¹. Together with my comment #2, I strongly recommend carefully re-investigating the morphology of the Pt species in 1Pt-SiO₂. I am feeling that there would be small Pt clusters rather than Pt single atoms or Pt oxide clusters in 1Pt-SiO₂.

5. Clarify whether the large Pt particles (Suppl. Fig. 2a) in 1Pt-SiO₂ were removed before CO oxidation tests. I am feeling that the sample with the large Pt particles was directly applied for CO oxidation (on page 12, lines 220~222). The catalyst performance test results could be biased.

6. Moreover, describe why the authors applied an extremely oxygen rich stream (20 vol. %) with quite high space velocity (134,000). Under such oxygen rich condition the light-off temperature could be lowered. Why did the authors perform stability tests under the much higher space velocity (300,000) and however, the stoichiometric condition (2%CO + 1%O₂).

7. Based on initial structural analysis results, the authors claimed that they successfully synthesized well-

dispersed alloy like clusters of Pt and Bi, largely attributed to the disintegrating and stabilizing power of Bi. However, the subsequent structural analysis results showed that Bi-stabilized Pt was aggregated into the large particles (also suggested by DRIFTS spectra) and the residual Bi formed a sort of BiO_x on the surface of Pt nanoparticles. If this is the case, I like to see high-resolution STEM images demonstrating the presence of the Bi species on the surface of Pt nanoparticles.

8. Discuss recent relevant findings on atomic precision ensemble optimization (O-metal-O), mostly reported in single atom catalyst communities. For example, *Science* 358, 1419-1423 (2007); 353 150-154 (2016), *Nat. Mater.* 18, 746-751 (2019), *Energy Environ. Sci.* 13, 1231-1239 (2020), *Nat. Commun.* 10:3803 (2019), *ACS Catal.* 9, 3978-3990 (2019).

Response to reviewer's comments

To Reviewer 1

The paper of Nan et al. presents the synthesis of a novel highly active CO oxidation catalyst based on Pt_xBi_yO_z clusters. The paper is very comprehensive, rich of profound characterization of the complex atomic structure of the catalyst and convincing in the interpretation of the data.

In a similar report recently published (Meunier F.C. et al. Angew. Chem. Int. Ed. 2020, 59,2–9) was already discussed the positive effect of oxidized Pt species and clearly showed the reaction mechanism of CO oxidation over these sites. Even though the presented paper shows very interesting results it does not reach unfortunately the high novelty standards of this journal, which need to fit to a broad audience.

Therefore, I cannot recommend the publication on Nature Communications and I encourage the authors to submit their manuscript to a more specific journal (e.g. ACS Catalysis). Also, I suggest to improve the introductory part highlighting the major novelty of their work. So far, this does not come through.

Author reply: We have read the article carefully entitled “Synergy between Metallic and Oxidized Pt Sites Unravelling during Room Temperature CO Oxidation on Pt/Ceria.” Meunier F.C. and co-workers reported that *both metallic and oxidized Pt are present in similar proportions under reaction conditions at the surface of ca.1 nm nanoparticles and found a synergy between these phases for metallic Pt phase providing strong adsorption sites for CO and oxidized Pt supplying reactive oxygen mainly according to the in-situ DRIFTS and NAP-XPS results.* In this work, our significant findings are as followings:

(1) **Formation of uniform Pt_xBi_yO_z binary metal oxide clusters to improve the thermal resistance to sintering of platinum on irreducible support (SiO₂).** The Bi-promoted Pt/SiO₂ catalysts with low content of platinum (0.8 wt.%) were prepared by incipient-wetness impregnation method and it exhibits excellent thermo-stability due to the formation of Pt_xBi_yO_z binary metal oxide cluster, as confirmed by the results of aberration-corrected HAADF-STEM, XAFS and in situ DRIFTS experiments. This binary oxide clusters could be maintained in small size (~2 nm) on the surface of inert support (SiO₂) after calcination at 550 °C in air. Meanwhile, our DFT results also

confirm the doping of Bi element could stabilize Pt atom and prevent the formation of huge Pt particles. As we all known, the inert oxides (such as SiO₂ and Al₂O₃) show poor ability to stabilize noble metal at subnanometer scale and frequent aggregation into huge particles due to weak interaction between metal and support. Other researcher has prepared silica-platinum catalysts with core-shell structure to prevent the aggregation of active metal, which always tend to hinder the contact of active site with reaction gas resulting in low catalytic activity (Kim et al. *ChemCatChem* 2019, 11, 4653–4659). We believe this strategy of synthesis could provide new guideline for preparation of small-size bimetal catalysts on inert support.

(2) **Our silica-supported platinum-bismuth catalysts catalyze CO oxidation efficiently with CO₂ production rate of 487 $\mu\text{mol}_{\text{CO}_2}\cdot\text{g}_{\text{Pt}}^{-1}\cdot\text{s}^{-1}$ at 110 °C and low apparent activation energy (~52 kJ/mol)**, which is much higher than that of Pt/CeO₂ (103 $\mu\text{mol}_{\text{CO}_2}\cdot\text{g}_{\text{Pt}}^{-1}\cdot\text{s}^{-1}$ at 130 °C, Nie et al. *Science* 358 (6369), 1419-1423). Thus, our Bi-promoted silica-supported platinum catalyst is a promising and high effective catalyst for CO oxidation. In another hand, the E_a value (~52 kJ/mol) of our PtBi-SiO₂ catalysts is much lower than that (~98 kJ/mol) of conventional silica-supported platinum catalyst.

(3) **Determination of new active site: metallic platinum cluster with surface partial oxidized Pt-[O]_x-Bi interface.** According to the comprehensive characterization results of *quasi* in-situ XAFS (new supplement in the revised manuscript), HAADF-STEM and XPS, we confirmed the change of platinum valance state and the structure evolution of active site. The results of *quasi* in-situ XANES indicate that the hydrogen reduction actually reduces the valance state of platinum species to meet the requirement of low oxide platinum species to catalyze CO oxidation (Ding et al. *Science* 2015, 350, 189–192) and the corresponding EXAFS results uncover the local coordination structure of active site with metallic Pt-Pt shell at 2.75 Å, Pt-O shell at 1.98 Å and Pt-O-Bi shell at 2.98 Å. Combination of the observation of cluster in HAADF-STEM images, the main phase of active site is metallic platinum cluster. In addition, the existence of Pt-O and Pt-O-Bi shell indicates that the Pt-[O]_x-Bi interface is on the surface of metallic platinum cluster.

(4) **This Pt-[O]_x-Bi interface could motivate CO oxidation through Mars-van Krevelen (MvK) mechanism, provide moderate CO adsorption and activate CO molecule with electron transformation between platinum atom and carbon monoxide.** The CO-TPR results in Figure 6b indicate that the Pt-[O]_x-Bi structure could provide abundant surface-active oxygen species,

which could react with CO molecule at ~ 50 °C through Mars-van Krevelen (MvK) mechanism rather than the conventional Langmuir-Hinshelwood mechanism for inert-supported metal catalysts. The Pt-CeO_x-TiO₂ interface reported by Yoo et al. (*Energy Environ. Sci.*, 2020, 13, 1231–1239) just provide the surface oxygen to react with CO molecule after 130 °C. In addition, we found an enormous difference in CO adsorption on active site. The in-situ DRIFTS results in Fig. 7 demonstrate that the formation of Pt-[O]_x-Bi structure could effectively prevent the excessive adsorption of CO on platinum site and there is a red-shift in CO adsorption frequency (~ 20 cm⁻¹) for 1Pt2Bi-SiO₂ due to more electron transporting from Pt atom in Pt-[O]_x-Bi structure to CO molecule to activate carbon monoxide as also confirmed by DFT results.

In conclusion, in our work, we provided a new strategy of synthesis for anti-sintered catalysts and identify the local coordination structure of Pt_xBi_yO_z binary metal oxide cluster. Furthermore, we discovered this PtBi-SiO₂ catalyst showing excellent catalytic activity in CO oxidation and determined a new active site: metallic platinum clusters with surface Pt-[O]_x-Bi structure. **It is significant to identify this Pt-[O]_x-Bi structure because this structure could provide superior active oxygen**, which may be suitable for many oxidation reactions catalyzed by platinum or bismuth catalysts, such as oxidation of 1,6-hexanediol, ethanol oxidation and methane oxidation. According to the reviewer's comment, we have made the careful modification about the introduction part to be more informative and persuasive (Page 3-4, Line 40-80).

The corrected description for introduction part in the revised manuscript: The CO oxidation reaction ($\text{CO} + 1/2 \text{O}_2 = \text{CO}_2$) is a well-known model reaction in heterogeneous catalysis, as well as a key step to resolve automobile exhaust containing CO, NO and hydrocarbons¹⁻⁵. According to the previous reports, the platinum (Pt)-based catalysts exhibit excellent catalytic activity in CO oxidation. In one hand, these high-performance catalysts always require reducible oxides as supports, such as CeO₂², FeO_x³, MnO₂⁴ and Co₃O₄⁴, due to their rich surface oxygen vacancy and the so-called “strong metal–support interaction”⁶⁻⁸. In another hand, the irreducible oxide (SiO₂ and Al₂O₃)-supported platinum catalysts with the advantages of commercial production, low cost and extensive application frequently show poor catalytic activity in CO oxidation especially in low temperature (< 150 °C) because of lack of surface activated oxygen and suitable active site^{9,10}. Meanwhile, SiO₂ shows inferior ability to stabilize active platinum species in small size (< 2 nm) consequently resulting in the deactivation of silica-supported platinum catalysts. Therefore, it is a

big challenge to prepare a kind of irreducible oxide-supported platinum catalyst with both excellent catalytic performance and thermo-stability to meet the demand of future exhaust-treatment system.

Many research groups have found that the addition of a secondary element, such as Sn⁵, K¹¹, Co¹² and Bi^{13,14}, distinctly improved the activity of silica- or alumina-supported platinum catalysts, no matter in the form of oxide clusters or metallic alloy for Pt. As for bismuth element, it has been widely used as the secondary element to improve the catalytic activity in various oxidative reactions^{13,14}, due to providing high content of mobile oxygen¹³, preventing the overoxidation of noble metal¹⁵ and suppressing adsorption of poisoning species¹⁶. Mondelli and co-workers has also confirmed that addition of Bi actually maintain Pt in a metallic state with the help of in-situ X-ray absorption spectroscopy (XAS)¹⁵. Meanwhile, Ding et al also reported metallic Pt nanoparticles show activity for CO oxidation¹⁷. However, the precise determination of active site (alloy or oxide solid solution?) and reaction mechanism (Langmuir-Hinshelwood or Mars-van Krevelen mechanism?) of Bi-promoted platinum catalysts are still in huge arguments^{14,18}. So, it is significant to prepare silica-supported platinum-bismuth catalysts to realize the high efficiency catalysis of CO oxidation and employ the comprehensive characterization methods to investigate the precise local structure of active site.

Moreover, many researchers have identified that various interfaces in platinum-based catalysts play a key role in in many industrially important reactions, such as metal–support^{3,19,20}, metal–oxide²¹⁻²³ and metal–metal hydroxide²⁴. Chen et al. reported iron nickel hydroxide-platinum nanoparticles (Pt-OH-Fe/Ni) were highly efficient for CO oxidation owing to abundant sites of Pt-OH-M interfaces²⁴. According to previous reports^{25,26}, it is easy to build multifarious atomic interface between metal and reducible oxide to improve catalytic performance. However, it is extremely difficult to structure effective metal-support or metal-oxide interface due to the infertile oxygen, poor reducibility and over-stable surface composition of irreducible oxide (SiO₂ and Al₂O₃). Therefore, it is great research interests to build stable and efficient interfaces on inert support to catalyze all kinds of heterogeneous reactions.

To Reviewer 2

1. *Many figures are not clear. The text in the figures is not clearly visible (For example, Fig. 4d).*

Authors need to improve the quality of figures to improve the readability of data.

Author reply: we thank the reviewer's suggestion. In the revised, we have replaced all figures with higher resolution pictures.

2. *Recent studies suggest that the synergistic catalytic effect on bimetal catalysts is related to the formation of reactive metal-oxide interfacial structure based on several operando surface characterization [For example, C. H Wu et al. Nature Catalysis 2, 78 (2019); T. S. Kim et al. ACS Catalysis 10, 10459 (2020); H. Lee et al. Nature Communications 9, 2235 (2018)]. Operando surface characterization such as ambient pressure XPS or environmental TEM would be ideal for identifying the reaction intermediates on PtBi nanoclusters under the reaction conditions. This point and relevant works can be discussed.*

Author reply: we thank the reviewer's suggestion. We agree with the reviewer's viewpoint: "Recent studies suggest that the synergistic catalytic effect on bimetal catalysts is related to the formation of reactive metal-oxide interfacial structure based on several operando surface characterization." Ambient pressure XPS and environmental TEM actually could provide the valence state and morphology information of active site. However, it is difficult to obtain enough good signal of platinum due to low content of platinum (0.9 wt.%) in a commercial NAP-XPS (*Angew. Chem. Int. Ed.* 2021, 60, 3799–3805, 5 wt.% Pt/CeO₂). Even the ex-situ XPS experiment showed not high signal-to-noise level in our work (see Fig. R1). Furthermore, the environmental TEM (ETEM) technique is difficult to acquire the atomic-resolution HAADF-STEM images of Pt-O-Bi structure or reaction intermediates in amorphous PtBi nanoclusters due to the similar atomic number (Z) between Pt (78) and Bi (83). In the other hand, the XAFS technique not only can observe the change of valence state of active metal, but also determine the local structure of reaction intermediates. It also can distinguish the signal of platinum and bismuth element well because of their specific absorption edges. More and more research groups have employed *quasi* in-situ (Xu et al. *Angew. Chem. Int. Ed.* 2020, 59, 21736–21744) or in-situ XAFS (Hao et al. *Nat. Catal.* 2019, 2, 448–456; Guo et al. *Topics in Catal.* 2009, 52, 1517–1524.) experiment to study the variation of valence state

and active site structure in different reaction. We have tried to carried out the in-situ XAFS experiment to detect the local structure of Pt species. However, we only acquired a bad signal-noise ratio XAFS spectrum due to the low content of platinum (0.8 wt. %) (*Phys. Chem. Chem. Phys.* 2013, 15, 18827–18834) and using powder sample. In another hand, we can press powder sample into pellet to improve the signal-noise ratio of in-situ XAFS, but only a few of Pt species on the surface of pellet can be reduced in hydrogen or participate in CO oxidation. It is difficult to acquire the real and reliable local structure of whole platinum species with in-situ XAFS experiment in this work. On the contrary, for *quasi* in-situ XAFS experiments, we can pretreat catalysts in powder state to acquire real active structure under different gas condition in a stainless reactor with two globe valves and then press powder into solid pellet to guarantee the quality of XAFS signal. Therefore, the *quasi* in-situ XAFS both can monitor the real active structure and guarantee the signal-noise ratio of XAFS spectrum. Before *quasi* in situ XAFS experiment, the sample was pretreated in a stainless reactor with two globe valves and tableting in glove box under nitrogen atmosphere (see Fig. R2) to isolate air and keeping active site structure constant. In the revised manuscript, we have added the *quasi* in situ XAFS results in Fig. 5 and the corresponding description in Page 14-16, Line 244-294 and the above literatures as Ref. 33-35.

Fig. R1 The Pt 4f XPS profiles of Pt/PtBi-SiO₂ samples.

Fig. R2 The reactor with two globe valves and glove box under nitrogen atmosphere

Fig. 5 The local coordination structure of Pt/PtBi-SiO₂ catalysts. *Quasi* in-situ Pt L₃-edge (a,c) XANES profiles and (b,d) EXAFS fitting results in *R* space for 1Pt-SiO₂ (a,b) and 1Pt2Bi-SiO₂ (c,d) under specific conditions.

The added description in the revised manuscript for Fig. 5:

According to the catalytic performance in CO oxidation with hydrogen reduction pretreatment, an obvious structural evolution occurred in Bi-promoted catalysts. Recently, more and more reports indicate that in-situ or *quasi* in-situ techniques could seize the evolution of active site and the synergistic effect on bimetal catalysts³³⁻³⁶. The *quasi* in situ XAFS experiment could acquire a good signal-noise ratio of XAFS spectrum for low content of metal and retain the real structure of active site under different condition. In order to elucidate the real active site structure in reductive and CO oxidation atmosphere, XAFS spectrums of 1Pt-SiO₂ and 1Pt2Bi-SiO₂ were collected after hydrogen

reduction at different temperature (150 and 210 °C) and 1 h of time-on-stream in CO oxidation (100 °C) in a stainless reactor with two globe valves and tableting in glove box under nitrogen atmosphere and ambient temperature for further XAFS experiments without exposure to air.

Combination with the catalytic performance in Fig. 3a and previous reported^{29,37}, the lower oxidized state of Pt species is appropriate for lower temperature CO oxidation. XANES data in Fig. 5a,c indicated that the valance state of platinum species is decreasing (+1.2 to +0.2 and +2.5 to +0.4 for 1Pt-SiO₂ and 1Pt2Bi-SiO₂ respective) as the increasing of hydrogen reduction temperature (150 to 210 °C) (Supplementary Table 5). The corresponding XANES profiles during CO oxidation following the hydrogen reduction at 210 °C were collected in Fig. 5a,c, the average valance of platinum slightly increases from +0.2 to +1.0 and +0.4 to +1.3 for 1Pt-SiO₂ and 1Pt2Bi-SiO₂ respective compared with that in reduced state at 210 °C. It demonstrated that the oxygen-rich reaction gas could make platinum species slight oxidative. Meanwhile, the similar valance of platinum species about +1 indicates that the difference in CO oxidation activity for 1Pt-SiO₂ and 1Pt2Bi-SiO₂ is not mainly due to platinum valance. Furthermore, the relevant EXAFS profiles were exhibited in Fig. 5c,d. For 1Pt-SiO₂, after hydrogen reduction, a main metallic Pt-Pt shell ($R \approx 2.75$ Å, $CN \approx 7.0-9.2$) was acquired and only a minor Pt-O shell ($R \approx 2.00$ Å, $CN \approx 0.6$) can be fitted at lower reduction temperature (150 °C), which may result in low activity due to no surface-active oxygen to participate in CO oxidation³⁸. For 1Pt2Bi-SiO₂, in order to require more reliable local coordination structure, we conducted the EXAFS fitting process with or without Pt-O-Bi shell in Supplementary Fig. 13. Obviously, Pt-O-Bi shell is an essential composition to acquire the most reasonable fitting results. In Fig. 5d and Supplementary Table 5, there is an obvious difference in local coordination structure compared with 1Pt-SiO₂. When the reduction temperature is at 150 °C, only partial platinum species is reduced to metallic Pt with a small $CN \approx 4.8$ at $R \approx 2.75$ Å, plus strong Pt-O ($R \approx 2.00$ Å, $CN \approx 2.1$) and Pt-O-Bi ($R \approx 2.98$ Å, $CN \approx 1.8$) shells. It proves the CO conversion platform of 1Pt2Bi-SiO₂ in Fig. 4a is due to incomplete evolution of active site at 150 °C. Furthermore, there is similar coordination structure for 1Pt2Bi-SiO₂-210 °C and 1Pt2Bi-SiO₂-CO oxidation, indicating the reduction temperature at 210 °C is appropriate for construction of optimal active sites. The EXAFS fitting results for Pt-Pt shell with $CN \approx 7.4$ also confirmed the average grain size of platinum cluster was ~2 nm for 1Pt2Bi-SiO₂³⁹ as observed in HAADF-STEM images (Fig. 4a-c). In addition, we found that the Pt-O ($R \approx 2.00$ Å, $CN \approx 1.5$) of 1Pt2Bi-SiO₂-CO oxidation

is higher than that ($R \approx 1.98 \text{ \AA}$, $CN \approx 0.6$) of 1Pt2Bi-SiO₂-210 °C, may due to the formation of more Pt-[O]_x-Bi structure in oxidative atmosphere. And the detection of Pt-O-Bi ($R \approx 3.00 \text{ \AA}$, $CN \approx 2.3$) composition (Fig. 5d, Supplementary Table 3) also validates the existence of Pt-[O]_x-Bi structure.

3. My main question is regarding Pt-O-Bi structure. H2-TPR data shown in Figure 5a shows the presence of BiO_x. In addition, XPS data shown in Figure 4i shows the moderate oxidation of Bi. Therefore, it can be assumed that Pt coexists with BiO_x clusters, and the metal-oxide interfaces could be reactive species. DFT calculation shows that Pt-O-Bi structure is the active site for CO oxidation. However, I don't see clear evidence of the formation of Pt-O-Bi structure.

Author reply: we thank the reviewer's comment. As shown in Fig. 6a (in the revised), there is an obvious peak shifting to higher temperature (97 to 162 °C) and as the bismuth doping content further increasing, the reduction peak shifted to 197 °C for 1Pt5Bi-SiO₂ in Supplement Fig. S15a. It indicates that the formation of strong interaction between platinum and bismuth species (Lan et al. *Mol. Catal.* 432, 23–30 (2017)). Furthermore, we acquire the local coordination structure of active site for PtBi-SiO₂ catalysts with Pt-O-Bi shell ($R = 3.03 \pm 0.02 \text{ \AA}$, $CN = 3.0 \pm 0.7$). In addition, fitting XAFS data with Pt-O-Pt shell cannot acquire reasonable result (negative coordination number) during the data-analysis process. Therefore, we determined that the oxide clusters in Bi-promoted samples are Pt_xBi_yO_z clusters. In addition, there is another reduction peak in Fig. 6a (in the revised) for fresh 1Pt2Bi-SiO₂ centered at 350 °C, which can be attributed to isolated BiO_x cluster or Bi³⁺ single atoms on the surface of silica due to excessive dopant of bismuth. This part of bismuth species has no interaction with platinum species and this reduction peak is similar to results of Bi₂O₃ in ref. 13.

After CO oxidation, the low oxidation state Pt^{δ+}-O-Bi structure ($0 < \delta < +2$) is formed due to hydrogen reduction. As shown in Fig. 6a (in the revised), there is a weak reduction peak at ~250 °C, which is attributed to the BiO_x clusters adjoining to platinum cluster. However, the oxygen provided by this BiO_x cluster only can be activated at high temperature ($> 200 \text{ °C}$), which is quite inconsistent with the low temperature CO oxidation activity. Meanwhile, the EXAFS fitting results of used 1Pt2Bi-SiO₂ in Fig. 5d (in the revised) confirm the existence of Pt-O-Bi structure with $R = 2.99 \pm 0.02$, $CN = 2.2 \pm 0.8$. Therefore, the surface Pt-[O]_x-Bi interface is the main active site for CO oxidation rather the BiO_x cluster around platinum cluster. We have corrected the corresponding description to

explain the reduction peak at 250 and 350 °C in Page 18, Line 334-339 in the revised version: “In addition, there are two reduction peaks at 250 and 350 °C, which are attributed to the high dispersion BiO_x cluster adjoining to platinum cluster and isolated BiO_x deposited on the surface of SiO₂. However, the oxygen provided by these BiO_x clusters only can be activated at high temperature (> 200 °C), which makes few contributions to low temperature CO oxidation activity.”

4. Figure 1 shows high-resolution TEM images and EDS mapping of elements of PtBi nanoclusters. I am wondering if PtBi is core-shell-like structure because of the surface segregation of one element. After the reaction, how does EDS elemental mapping change?

Author reply: We thank the reviewer’s comment. We have carried out the HAADF-STEM images and the corresponding EDS mapping of single oxide clusters for fresh 1Pt2Bi-SiO₂. These experimental findings demonstrate the uniform dispersion of platinum and bismuth element rather than core-shell structure (Fig. 1c). After the CO oxidation, it is also found that the platinum and bismuth elements are distributed uniformly, according to the STEM-EDS mapping results for individual cluster in Fig. 4d. The elemental mapping in the previous manuscript is not clear because of the demagnification of the image. In the revised manuscript, the enlarged mapping images are used to display clearly the elemental distribution (Fig. 4d). Due to the amorphous nature of PtBi nanocluster as well as similar atomic number (Z) of Pt and Bi, it is almost not able to identify the core-shell structure in TEM/STEM images. However, the EXAFS fitting results (Fig. 5d) can confirm the existence of Pt-[O]_x-Bi structure (Pt-O-Bi shell) and the 2043 cm⁻¹ of CO adsorption on Pt-[O]_x-Bi site in in-situ DRIFTS experiments (Fig. 7) also indicate that the Pt-[O]_x-Bi structure is on the surface of platinum cluster. In the revised manuscript, we have added the EDS mapping images of fresh 1Pt2Bi-SiO₂ in supplementary Fig. 3c and changed the EDS mapping images of used 1Pt2Bi-SiO₂ in Fig. 4d and corrected the corresponding description in Page 7, Line 120-122: “When the EDS mapping was conducted for the individual cluster, no obvious core-shell structure can be observed (Supplementary Fig. 3c).” and Page 13, Line 231-233: “Meanwhile, the related STEM-EDS mapping results of cluster indicated that the Pt and Bi elements still distribute together, not core-shell structure at the same area (Fig. 4d and Supplementary Fig. 11).”

Fig. 4d the corresponding EDS element mapping of used 1Pt2Bi-SiO₂

To Reviewer 3:

1. Introduction should be revised to be more informative and persuasive. Overall, introduction of the current manuscript is not persuasive and lack of important information, making the audience difficult to understand the urgency and the scientific value of this work.

Several suggestions:

1-1) Deduce and present the motivation that drove the authors to design Pt-based catalysts supported on SiO₂ for CO oxidation by discussing more precisely the advantages/disadvantages of using irreducible oxides as a supporting material for Pt. It is well known that Pt nanoparticles supported on such irreducible oxides catalyze CO oxidation through the Langmuir-Hinshelwood mechanism, rather than the interface-mediated Mars-van Krevelen mechanism.

1-2) Present scientific reason of using Bi as a dopant.

1-3) In the 3rd paragraph, provide in-depth discussion on the relevant recent studies on the catalytic role of various interfaces. Refer to Nat. Catal. 2, 955-970 (2019) and ACS Catal. 8, 7368-7387 (2018) and references therein.

Author reply: we thank the reviewer's suggestion about the introduction part. In the revised, we have recomposed the introduction part in Page 3-4 and added the above literatures as ref. 25 and 26.

The added description in the revised manuscript for introduction part: The CO oxidation reaction ($\text{CO} + 1/2 \text{O}_2 = \text{CO}_2$) is a well-known model reaction in heterogeneous catalysis, as well as a key step to resolve automobile exhaust containing CO, NO and hydrocarbons¹⁻⁵. According to the previous reports, the platinum (Pt)-based catalysts exhibit excellent catalytic activity in CO oxidation. In one hand, these high-performance catalysts always require reducible oxides as supports, such as CeO₂², FeO_x³, MnO₂⁴ and Co₃O₄⁴, due to their rich surface oxygen vacancy and the so-called "strong metal-support interaction"⁶⁻⁸. In another hand, the irreducible oxide (SiO₂ and Al₂O₃)-supported platinum catalysts with the advantages of commercial production, low cost and extensive application frequently show poor catalytic activity in CO oxidation especially in low temperature (<150 °C) because of lack of surface activated oxygen and suitable active site^{9,10}. Meanwhile, SiO₂ shows inferior ability to stabilize active platinum species in small size (< 2 nm) consequently resulting in the deactivation of silica-supported platinum catalysts. Therefore, it is a big challenge to prepare a kind of irreducible oxide-supported platinum catalyst with both excellent

catalytic performance and thermo-stability to meet the demand of future exhaust-treatment system.

Many research groups have found that the addition of a secondary element, such as Sn⁵, K¹¹, Co¹² and Bi^{13,14}, distinctly improved the activity of silica- or alumina-supported platinum catalysts, no matter in the form of oxide clusters or metallic alloy for Pt. As for bismuth element, it has been widely used as the secondary element to improve the catalytic activity in various oxidative reactions^{13,14}, due to providing high content of mobile oxygen¹³, preventing the overoxidation of noble metal¹⁵ and suppressing adsorption of poisoning species¹⁶. Mondelli and co-workers has also confirmed that addition of Bi actually maintain Pt in a metallic state with the help of in-situ X-ray absorption spectroscopy (XAS)¹⁵. Meanwhile, Ding et al also reported metallic Pt nanoparticles show activity for CO oxidation¹⁷. However, the precise determination of active site (alloy or oxide solid solution?) and reaction mechanism (Langmuir-Hinshelwood or Mars-van Krevelen mechanism?) of Bi-promoted platinum catalysts are still in huge arguments^{14,18}. So, it is significant to prepare silica-supported platinum-bismuth catalysts to realize the high efficiency catalysis of CO oxidation and employ the comprehensive characterization methods to investigate the precise local structure of active site.

Moreover, many researchers have identified that various interfaces in platinum-based catalysts play a key role in in many industrially important reactions, such as metal–support^{3,19,20}, metal–oxide²¹⁻²³ and metal–metal hydroxide²⁴. Chen et al. reported iron nickel hydroxide-platinum nanoparticles (Pt-OH-Fe/Ni) were highly efficient for CO oxidation owing to abundant sites of Pt-OH-M interfaces²⁴. According to previous reports^{25,26}, it is easy to build multifarious atomic interface between metal and reducible oxide to improve catalytic performance. However, it is extremely difficult to structure effective metal-support or metal-oxide interface due to the infertile oxygen, poor reducibility and stable surface composition of irreducible oxide (SiO₂ and Al₂O₃). Therefore, it is great research interests to build stable and efficient interfaces on inert support to catalyze all kinds of heterogeneous reactions.

2. I see that large Pt particles were developed during calcination (also evidence by the XRD spectrum) in 1Pt-SiO₂. Presumably, the most Pt-like XANES profile of 1Pt-SiO₂ compared with the others containing Bi suggests that Pt particles were formed in 1Pt-SiO₂. Based on their EXAFS fitting profile result, the authors stated that platinum oxide clusters (line 108, on page 6) were

formed together with large Pt particles in 1Pt-SiO₂. However, I note that the XAS spectra of small Pt species (clusters) or Pt single atoms, which make close and strong contact between Pt and the oxygen ion of the supporting oxide, could be largely shifted toward that of PtO₂. Therefore, it is cautious to state that Pt oxide clusters (rather than single atoms or small sub-nanometer sized Pt clusters) were formed on SiO₂.

Author reply: We thank the reviewer's comment. According to previous reports, the platinum valance state is determined by the intensity of white peak (*Langmuir* 2001, 17, 3047-3050) rather than the shift of adsorption edge in XANES profiles. The average valance state of platinum is +1.8 (supplementary Table 2), according to the linear combination fitting of XANES profiles, about 45% in Pt⁰ and 55% in Pt⁴⁺ (Fig. R3). Furthermore, our previous report (*J. Phys. Chem. C* 2017, 121, 25805–25817) have identified that the fraction of different Pt species could be calculated by coordination number (as shown in Table R1) about 50% in cluster and 50% in particle based on the CN of metallic Pt–Pt and Pt–O–Pt shell. Therefore, combination of ratio of cluster (50%) and Pt⁴⁺ (55%), we can confirm that the small size cluster is almost platinum oxide cluster rather than Pt⁰ cluster. Besides, as a reference, we also prepared a sample called 1Pt-SiO₂-400, which was calcinated at 400 °C under air without huge Pt particle and the same synthetic method with 1Pt-SiO₂. For 1Pt-SiO₂-400, the XRD (Supplementary Fig. 7c), TEM (Supplementary Fig. 7a,b) and XANES (Fig. R4, 100% in Pt⁴⁺) results show that the small-size platinum species is totally in the form of Pt_xO₂ oxide clusters rather than Pt⁰ cluster after calcination in air. Therefore, as the increase of calcination temperature, the sintering of platinum species results in the generation of huge metallic platinum particles and according to the previous reports (Jones et al. *Science* 2016, 353, 150-154; Zhai et al. *Science* 2010, 329, 1633-1636; Ke et al. *ACS Catal.* 2015, 5, 5164-5173), no matter on the reducible (CeO₂) or irreducible (SiO₂, Al₂O₃) oxide supports, the small-size platinum species is still in the form of Pt_xO₂ oxide clusters or Pt^{δ+} single atom after calcination under air. Hence, we can conclude that the small-size Pt_xO₂ oxide clusters and huge metallic platinum particles simultaneously exist in fresh 1Pt-SiO₂.

Fig. R3 the linear combination fitting results of XANES for 1Pt-SiO₂.

Fig. R4 the linear combination fitting results of XANES for 1Pt-SiO₂-400.

Table R1. Calculation on the efficiencies of different platinum species for 1Pt-SiO₂ samples.

	Atom	Cluster	Particle
Occupation(O)	$(CN_{\text{Pt-O}} - CN_{\text{Pt-O-Pt}})/8$	$CN_{\text{Pt-O-Pt}}/8$	$CN_{\text{Pt-Pt}}/12$
Fraction (f)	$O_{\text{atom}}/(O_{\text{atom}} + O_{\text{cluster}} + O_{\text{particle}})$	$O_{\text{cluster}}/(O_{\text{atom}} + O_{\text{cluster}} + O_{\text{particle}})$	$O_{\text{particle}}/(O_{\text{atom}} + O_{\text{cluster}} + O_{\text{particle}})$

Moreover, it is highly likely that the Pt particles presented in Figure S2a were not physically separated from the sample before conducting further analyses. If this is the case, the authors may centrifuge the sample and collect the inherent XAS spectra of the small Pt species supported on SiO₂, which are not biased from Pt particles.

Author reply: We thank the reviewer's suggestion. We have conducted the centrifuged experiment for 1Pt-SiO₂ at 10,000 r/min for 60 mins with TG16-WS table-top high-speed centrifuge (Shanghai Luxiangyi Co. Ltd.). The XRD result (see Fig. R5) shows that the huge Pt particles cannot be physically separated from 1Pt-SiO₂. In another hand, the EXAFS profile of 1Pt-SiO₂-400 without

huge Pt particles in supplement Fig. 14a shows only Pt-O and Pt-O-Pt shells.

Fig. R5 the XRD profile of 1Pt-SiO₂ after centrifugation

Supplementary Fig. 14 Pt L₃-edge EXAFS fitting results in *R* space for fresh 1Pt-SiO₂-400.

3. The authors showed that Bi phenomenologically prevents formation of Pt particles. I agree with this statement. However, additional discussion from thermodynamic or energetic perspectives (highly likely based on DFT calculation results) on how Bi disintegrates or stabilizes Pt should be provided.

Author reply: we thank the reviewer's suggestion. Following the reviewer's suggestion, we have performed supplementary DFT calculations and further analyzed the presence status of Pt from an energy point of view. We use $\Delta E_{average}$ as an indicator to describe the degree to which the Pt species is stabilized under different conditions:

$$\Delta E_{average} = [E(Pt_x Bi_y O_z) - (x \cdot E(Pt) + y \cdot E(Bi) + z \cdot E(O_2)/2)]/x,$$

Where $E(Pt_x Bi_y O_z)$ is the total energy of the whole system, $E(Pt)$, $E(Bi)$, and $E(O_2)$ is the energy of a Pt atom, a Bi atom, and an O₂ molecule, respectively. It is worth noting that here the denominator only contains the number of Pt atoms, because we regard Pt as the object of existence,

while O₂ and Bi are merely the environment in which the Pt species exists.

We have considered three systems, Pt₅₅@PtO_x-10Bi, Pt₅₅@PtO_x, and Pt₅₅, for the comparison of the $\Delta E_{average}$ value. Here, the PtO_x shell contains 32 Pt atoms and 66 O atoms in the simulation model. Our results show that the corresponding value for Pt₅₅@PtO_x-10Bi, Pt₅₅@PtO_x, and Pt₅₅ is calculated to be -6.11 eV, -5.46 eV, and -4.52 eV, respectively. The large difference in the energy value per Pt atom indicates that the Pt atoms are stabilized the most when the O₂ atmosphere and the Bi dopants both exist. It thus supports the conclusion from the experiments that Bi can prevent the formation of pure Pt particles, as in this case, the Pt species will form bonds with O and Bi atoms at the same time. In the revised manuscript, we have added the corresponding description in Page 7, Line 131-135: “Therefore, the HAADF-STEM and XRD results directly indicated that bismuth oxide species as a promoter could phenomenologically prevent the formation of pure huge Pt particles (50–100 nm) on an inert support (SiO₂), which is further supported by our density functional theory calculations with more details in the Supplementary Table 2.”

Supplementary Table 2: The calculation formula for energy of Pt species stabilizing under different conditions.

$$\Delta E_{average} = [E(Pt_x Bi_y O_z) - (x \cdot E(Pt) + y \cdot E(Bi) + z \cdot E(O_2)/2)]/x$$

4. The DRIFTS spectra of 1Pt-SiO₂ presented in Fig. 2 and Suppl. Fig. 6 are needed to be more carefully analyzed. The location of the CO-probe peak is dependent to the supporting material and pre- or post-treatment condition. I think the peak at 2075 cm⁻¹ is somewhat originated from metallic Pt species, and the peak centered at 2093 cm⁻¹ could be the one from the weakly (positively) charged ionic Pt of small Pt clusters, rather than one from Pt single atoms. Note that the ref. 23 reports the IR spectra of Pt supported on FeO_x. To the best of my knowledge, the CO-probe peaks of Pt single atoms supported on oxides usually appear at above 2100 cm⁻¹. Together with my comment #2, I strongly recommend carefully re-investigating the morphology of the Pt species in 1Pt-SiO₂. I am feeling that there would be small Pt clusters rather than Pt single atoms or Pt oxide clusters in 1Pt-SiO₂.

Author reply: We thank the reviewer's comment and suggestion. First, the DRIFTS spectra in Fig. 2 and Supplementary Fig. 6 are attributed to the comparison of CO adsorption of 1Pt-SiO₂-400 and 1Pt2Bi-SiO₂. The reason why we compare the CO adsorption of 1Pt2Bi-SiO₂ with 1Pt-SiO₂-400 rather than 1Pt-SiO₂ is to exclude the size effect of active site on DRIFTS spectra. The average size of oxide cluster is 1.7±0.4 nm for 1Pt2Bi-SiO₂ and 1.8±0.3 nm for 1Pt-SiO₂-400. As our reply in comment 2, only Pt_xO_z clusters and some Pt^{δ+} single atom exist in 1Pt-SiO₂-400 without metallic platinum particles. Therefore, the two CO adsorption peaks centered at 2093 and 2075 cm⁻¹ must be attributed to the CO molecule adsorbed on Pt^{δ+} single atom and Pt_xO_z clusters. Meanwhile, we agree with the reviewer's viewpoint: "*The location of the CO-probe peak is dependent to the supporting material and pre- or post-treatment condition.*" In addition, the collection temperature of spectra also effects the CO adsorption frequency. Therefore, according to the article "Ding et al. *Science*, 350 (6257), 189-192", their 0.6 wt.% Pt-SiO₂ (Al doped) possesses similar platinum loading (0.6 wt.%), species composition (Pt single atoms and nanoparticles) and collection temperature (100 °C). In figure S8 of Ding's work (Ding et al. *Science*, 350 (6257), 189-192), the band at ~2092 cm⁻¹ is attributed to CO molecule adsorbed on Pt^{δ+} single atom, which is identical with the band at 2093 cm⁻¹ in our work. In addition, Ding et al. also reported that the CO adsorption peak on oxide Pt nanoparticles (XPS results in Figure S7) is between 2050 and 2080 cm⁻¹ (Figure 2A) and we have confirmed that the CO adsorption on Pt⁰ particle is at 2060 cm⁻¹ in Fig. 7a of our work. Therefore, we can attribute the CO adsorption band at 2075 cm⁻¹ to CO adsorbing on platinum oxide clusters. Therefore, in the revised manuscript, we have replaced ref.23 by "Ding et al. *Science*, 350 (6257), 189-192".

5. Clarify whether the large Pt particles (Suppl. Fig. 2a) in 1Pt-SiO₂ were removed before CO oxidation tests. I am feeling that the sample with the large Pt particles was directly applied for CO oxidation (on page 12, lines 220~222). The catalyst performance test results could be biased.

Author reply: we thank the reviewer's suggestion. We have considered the effect of huge Pt particles on CO oxidation activity. Therefore, we compared the CO oxidation activity of 1Pt-SiO₂ (with huge Pt particles about 50-100 nm) and 1Pt-SiO₂-400 (only Pt_xO_z clusters without huge Pt particles) in supplementary Fig. 9b, in which no obvious difference in CO oxidation activity can be observed. Thus, we think size-effect is not the main reason for the difference between Bi-free and

Bi-promoted catalysts. It further indicates that the promotion of activity is depended on the formation of new active site (Pt-O-Bi structure) rather than the size effect. In the revised manuscript, we have added the corresponding description in Page 11, Line 206-209: “For eliminating size-effect on active site, 1Pt-SiO₂-400 was similarly inactive for CO oxidation at 100 °C reaction with complete conversion of CO at 200 °C (Supplementary Fig. 9b), though possessing similar cluster size (1.8±0.1 nm) to 1Pt2Bi-SiO₂ (Supplementary Fig. 7).”.

Supplementary Fig. 9 (b) the catalytic performance of 1Pt-SiO₂ and 1Pt-SiO₂-400.

6. Moreover, describe why the authors applied an extremely oxygen rich stream (20 vol. %) with quite high space velocity (134,000). Under such oxygen rich condition the light-off temperature could be lowered. Why did the authors perform stability tests under the much higher space velocity (300,000) and however, the stoichiometric condition (2%CO + 1%O₂).

Author reply: we thank the reviewer’s comment. According to the precious reports (Min et al. *Chem. Rev.* **2007**, 107, 2709-2724; Patrick et al. *Topics in Catal.* **2004**, 30-31, 1 - 4; Nie et al. *Science* **2017**, 358, 1419–1423; Wang et al. *Catalysis Reviews: Science and Engineering*, **2015**, 57:1, 79-144; Binder et al. *Angew. Chem. Int. Ed.* **2015**, 54, 13263 - 13267), the oxidation of CO has been extensively studied to control the criteria pollutants (CO) in automotive emission abatement. In one hand, the irreducible oxide-supported platinum catalysts follow the Langmuir–Hinshelwood (LH) mechanism and the oxygen-rich reaction gas is beneficial to exhibit excellent catalytic performance in CO oxidation. In another hand, to mimic lean-burn diesel engine exhaust (excess O₂ in the reactant, CO/O₂ = 1/25 in molar ratio, Nie et al. *Science* **2017**, 358, 1419–1423) and match standard vehicle exhaust conditions with high gas hourly space velocity, we employed oxygen-rich reaction

gas (1% CO/20% O₂/N₂) and high gas space velocity (134,000 mL g_{cat}⁻¹ h⁻¹). Meanwhile, we also carried out the stability test of 1Pt2Bi-SiO₂ in Fig. 3d at 150 °C and 200,000 mL g⁻¹ h⁻¹ in 1% CO/20% O₂/N₂ reaction gas with CO conversion at 70–80%. Furthermore, we have studied that high temperature is crucial for operations under high engine loads (Nie et al. *Science* **2017**, 358, 1419–1423 in Figure S7A). However, we found that the CO conversion is always maintained at 100 % at 500 °C even with 3 mg 1Pt2Bi-SiO₂ catalyst, which cannot distinguish the excellent durability or excess mass of catalysts. Therefore, we employ the reaction gas at stoichiometric condition (2% CO + 1% O₂) with super high gas hourly space velocity (300,000 mL g_{cat}⁻¹ h⁻¹) to control the CO conversion at 95% in Supplementary Fig. 9c. Therefore, in the revised manuscript we have added the corresponding description in Page 10, Line 182-185: “To mimic lean-burn diesel engine exhaust and acquire the best catalytic performance, we used excess O₂ in the reactant (CO/O₂ = 1/20)²⁸. A gas hourly space velocity of 134,000 ml g_{cat}⁻¹ hour⁻¹ was tried to match standard vehicle exhaust conditions.” and Page 12, Line 215-219: “Additionally, 1Pt2Bi-SiO₂ showed remarkable thermo-stability for ~70 hours at 150 °C and 200,000 mL g⁻¹ h⁻¹ in 1% CO/20% O₂/N₂ (Fig. 3d) and catalytic stability at high temperature (500 °C) under the extremely high space velocity to maintain the CO conversion at 95% (300,000 mL·g_{cat}⁻¹·h⁻¹, Supplementary Fig. 9c).”

7. Based on initial structural analysis results, the authors claimed that they successfully synthesized well-dispersed alloy like clusters of Pt and Bi, largely attributed to the disintegrating and stabilizing power of Bi. However, the subsequent structural analysis results showed that Bi-stabilized Pt was aggregated into the large particles (also suggested by DRIFTS spectra) and the residual Bi formed a sort of BiO_x on the surface of Pt nanoparticles. If this is the case, I like to see high-resolution STEM images demonstrating the presence of the Bi species on the surface of Pt nanoparticles.

Author reply: we thank the reviewer’s comment. According to XRD, HAADF-STEM, and EXAFS results, the formation of Pt_xBi_yO_z binary oxide clusters can stabilize platinum species in small size (~1.5 nm). After CO oxidation, we found that Pt_xBi_yO_z binary oxide clusters transform into ~2.0 nm metallic Pt cluster with surface Pt-[O]_x-Bi structure according to the metallic Pt-Pt and Pt-O-Bi shells in EXAFS fitting results in 1Pt2Bi-SiO₂ (Fig. 5 and Supplementary Table 4) due to hydrogen pretreatment. No obvious aggregation of platinum species occurred in 1Pt2Bi-SiO₂ after CO oxidation. As discussed above, due to the amorphous nature of PtBi nanocluster as well as similar

atomic number (Z) of Pt (78) and Bi (83), it is almost impossible to distinguish Bi from Pt atoms, so it is difficult to demonstrate the presence of Bi species on the surface of Pt cluster from the atomic-resolution HAADF-STEM images. Here, the core-shell structure is confirmed by the EXAFS fitting and in situ DRIFTS results. In the revised manuscript, we have changed the EDS mapping images of used 1Pt2Bi-SiO₂ in Fig. 4d and made modification about description in Page 13, Line 231-233: “Meanwhile, the related STEM-EDS mapping results of cluster indicated that the Pt and Bi elements still distribute together at the same area (Fig. 4d and Supplementary Fig. 11).”

Fig. 4d the corresponding EDS element mapping of used 1Pt2Bi-SiO₂

8. Discuss recent relevant findings on atomic precision ensemble optimization (*O-metal-O*), mostly reported in single atom catalyst communities. For example, *Science* 358, 1419-1423 (2007); 353 150-154 (2016), *Nat. Mater.* 18, 746-751 (2019), *Energy Environ. Sci.* 13, 1231-1239 (2020), *Nat. Commun.* 10:3803 (2019), *ACS Catal.* 9, 3978-3990 (2019).

Author reply: we thank the reviewer’s suggestion for the discussion about the single atom catalysts. In the revised manuscript, we have added the discussion about single atom catalysts for introduction and in-situ DRIFTS part in Page 4, Line 74-80, and Page 20, Line 360-364 and the above literatures as ref. 25,26,30,46, as below:

According to previous reports^{25,26}, it is easy to build multifarious atomic interface between metal and reducible oxide to improve catalytic performance. On the contrary, it is extremely difficult to structure effective metal-support interface due to the infertile oxygen, poor reducibility and stable surface composition of irreducible oxide (SiO₂ and Al₂O₃). Therefore, it is great research interests to build stable and efficient interfaces to catalyze all kinds of heterogeneous reactions.

Because, recent reports have indicated that a lot of atomically dispersed platinum catalysts shows low catalytic activity in CO oxidation due to over strong adsorption of CO, even with

maximized number density of reaction sites^{25,26,30,46}. Therefore, it is significant to investigate the CO adsorption on active site for Bi-free and Bi-promoted catalysts.

REVIEWERS' COMMENTS

Reviewer #2 (Remarks to the Author):

I looked into the revised manuscript and authors' response letter carefully. I think the authors addressed my concerns properly, and now the presentation and quality of the paper have been improved significantly. Considering the importance and impact of this work, I suggest that the paper be accepted as it is.

Reviewer #3 (Remarks to the Author):

The authors have addressed appropriately my previous concerns. However, further polishing is required.

Comments:

1. Introduction part is needed to be more precisely revised. There are some logical flaws. For example, the last sentence of the first paragraph (newly added) claims that “it is a big challenge to prepare a kind of irreducible oxide-supported platinum catalyst with both excellent catalytic performance and thermo-stability to meet the demand of future exhaust-treatment system.” However, note that the above statement was not appropriately supported by the other sentences. The authors are required to present what is “the demand of future exhaust-treatment system” and why both excellent performance and stability should be achieved in IRREDUCIBLE oxide-supported Pt catalysts.

Moreover, the following sentences should be carefully revised.

“Moreover, many researchers have identified that various interfaces in platinum-based catalysts play a key role in in many industrially important reactions...”

“However, it is extremely difficult to structure effective metal-support or metal-oxide interface due to the infertile oxygen,...”

(No Verb)

“Therefore, it is great research interests to build stable and efficient interfaces on inert support to catalyze all kinds of heterogeneous reactions.”

(over-emphasized)

2. I understand that there are several relevant studies in the literature reporting the promotional effect of Bi on the catalytic performance of Pt for oxidation reactions. However, even though the reaction mechanism of oxidation reaction catalyzed by Pt-Bi catalysts and the location of the active site on Pt-Bi catalysts are under debate, such lack of a specific knowledge itself does not provide a scientific motivation for using Pt-Bi catalyst throughout this study.

I like to see the authors present a clear statement on the final goal of this study and the potential impact of this work on the relevant fields. The authors can secure the scientific motivation and value of this

study by emphasizing how the Pt-Bi alloy catalyst and the authors' approach align well with the grand challenge in the relevant fields. I think the authors may reorganize the 2nd and the 3rd paragraphs to be more informative and logically concrete.

3. There are still some broken sentences. Revise the entire manuscript carefully.

4. Supplementary table 2 could be more informative. Please provide the calculated DEaverage and the structural models together.

5. I read Supplementary Figure 9b differently from the authors. I see that the 1Pt-SiO₂-400 is less reactive for CO oxidation than 1Pt-SiO₂ at below 150 C. This could be originated from the following two cases: 1) different kinds of the reaction sites with different activation energy barriers are working in two systems or 2) 1Pt-SiO₂ provides dense active sites (sites/Pt used) than 1Pt-SiO₂-400. The authors may provide the site or mass normalized rate plots (TOF or TON).

Response to Reviewers

Reviewer #2 (Remarks to the Author):

I looked into the revised manuscript and authors' response letter carefully. I think the authors addressed my concerns properly, and now the presentation and quality of the paper have been improved significantly. Considering the importance and impact of this work, I suggest that the paper be accepted as it is.

Author Reply: We are very grateful to your encouraging and positive comments and really appreciate your agreement of acceptance with this revised manuscript.

Reviewer #3 (Remarks to the Author):

1. Introduction part is needed to be more precisely revised. There are some logical flaws. For example, the last sentence of the first paragraph (newly added) claims that “it is a big challenge to prepare a kind of irreducible oxide-supported platinum catalyst with both excellent catalytic performance and thermo-stability to meet the demand of future exhaust-treatment system.” However, note that the above statement was not appropriately supported by the other sentences. The authors are required to present what is “the demand of future exhaust-treatment system” and why both excellent performance and stability should be achieved in IRREDUCIBLE oxide-supported Pt catalysts.

Author reply: we thank reviewer's comment. According to previous report (Nie et al. *Science* **2017**, 358, 1419–1423), the demand of future exhaust-treatment system is achieving the 90% conversion of all criteria pollutants at 150 °C. For irreducible oxide-supported platinum catalysts, the weak interaction between irreducible support and active site frequently results in the aggregation of active metal and the deactivation of catalysts. In another hand, it is difficult to achieve excellent catalytic performance on

irreducible oxide-supported platinum due to improper active site and the lack of surface active-oxygen species. Therefore, it is a big challenge to improve thermo-stability and catalytic performance of inert-supported platinum catalysts. Actually, both reducible oxide-supported and irreducible oxide-supported platinum catalysts could meet the demand of future exhaust-treatment system. Thus, we have deleted “to meet the demand of future exhaust-treatment system.” in the sentence of “it is a big challenge to prepare a kind of irreducible oxide-supported platinum catalyst with both excellent catalytic performance and thermo-stability to meet the demand of future exhaust-treatment system.” in the revised version and only emphasized the significance of developing highly efficient and stable irreducible-supported platinum catalysts.

In the revised manuscript, we have added the corresponding descriptions in Page 3, Line 48-52: “The CO oxidation reaction ($\text{CO} + 1/2 \text{O}_2 = \text{CO}_2$) is a well-known model reaction in heterogeneous catalysis, as well as a key step to resolve automobile exhaust containing CO, NO and hydrocarbons¹⁻⁵. According to the previous reports, the platinum (Pt)-based catalysts exhibit excellent catalytic activity in CO oxidation. In one hand, these high-performance catalysts always require reducible oxides as supports, such as CeO_2 ², FeO_x ³, MnO_2 ⁴ and Co_3O_4 ⁴, due to their rich surface oxygen vacancy and the so-called “strong metal–support interaction”⁶⁻⁸. In another hand, the irreducible oxide (SiO_2 and Al_2O_3)-supported platinum catalysts with the advantages of commercial production, low cost and extensive application frequently show poor catalytic activity in CO oxidation especially in low temperature because of the lack of surface activated oxygen and suitable active site^{9,10}. Meanwhile, the aggregation of platinum species in SiO_2 -supported platinum catalysts frequently results in the deactivation of catalysts in CO oxidation. Therefore, it is a big challenge to develop a kind of irreducible oxide-supported platinum catalyst with both excellent catalytic performance in low temperature and thermo-stability for practical application.”

Moreover, the following sentences should be carefully revised.

*“Moreover, many researchers have identified that various interfaces in platinum-based catalysts play a key role **in in** many industrially important reactions...”*

Author reply: we thank reviewer's comment. In the revised manuscript, we have deleted the extra "in" of this sentence.

"However, it is extremely difficult to structure effective metal-support or metal-oxide interface due to the infertile oxygen,..."(No Verb)

Author reply: we thank reviewer's suggestion. We have added the missing verb "build" and replaced the sentence by "However, it is extremely difficult to build effective metal-support or metal-oxide interface due to the infertile oxygen,..."

"Therefore, it is great research interests to build stable and efficient interfaces on inert support to catalyze all kinds of heterogeneous reactions." (over-emphasized)

Author reply: we thank the reviewer's comment. In the revised manuscript, we have replaced this sentence by "Therefore, it is great research interests to build stable and efficient interfaces on inert support to catalyze CO oxidation reaction."

2. I understand that there are several relevant studies in the literature reporting the promotional effect of Bi on the catalytic performance of Pt for oxidation reactions. However, even though the reaction mechanism of oxidation reaction catalyzed by Pt-Bi catalysts and the location of the active site on Pt-Bi catalysts are under debate, such lack of a specific knowledge itself does not provide a scientific motivation for using Pt-Bi catalyst throughout this study.

I like to see the authors present a clear statement on the final goal of this study and the potential impact of this work on the relevant fields. The authors can secure the scientific motivation and value of this study by emphasizing how the Pt-Bi alloy catalyst and the authors' approach align well with the grand challenge in the relevant fields. I think the authors may reorganize the 2nd and the 3rd paragraphs to be more informative and logically concrete.

Author reply: we thank reviewer's suggestion about the introduction part. In the revised manuscript, we have reorganized the second paragraph to clearly state our final goal and potential impact of our work via combination of PtBi alloy and our approach.

In addition, the active site in our work is about Pt-[O]_x-Bi interface. So, we maintained the description about interface in the third paragraph.

In the revised manuscript, we have modified the introduction part in Page 4, Line 61-74: Many research groups have found that the addition of a secondary element, such as Sn⁵, K¹¹, Co¹² and Bi^{13,14}, distinctly improved the activity of silica- or alumina-supported platinum catalysts, no matter in the form of oxide clusters or metallic alloy for Pt. As for bismuth element, it has been widely used as the secondary element to improve the catalytic activity in various oxidative reactions^{13,14}, due to providing high content of mobile oxygen¹³, preventing the overoxidation of noble metal¹⁵ and suppressing adsorption of poisoning species¹⁶. Many research groups have prepared crystalline platinum-bismuth alloy to promote the activity of some oxidation reactions. However, bismuth as an oxyphilic element is easily to form Bi³⁺ species in oxidative atmosphere resulting in oxidation of platinum-bismuth alloy. Feng et al. reported that the metallic and positive bismuth species coexists in Pt-Bi/SBA-15 catalysts for the selective oxidation of glycerol¹⁷. In addition, the deposit location of bismuth species (on the surface of support or active site) also have huge influence on catalytic performance^{17,18}. Therefore, it is significant for us to determine the precise local structure of active site (alloy or oxide cluster?¹⁴) and the practical valance state of bismuth and platinum (metallic or positive charge?^{19,20}) for Bi-promoted platinum catalysts in oxidation reactions. In this work, we provide a special viewpoint of active site: partially oxidized Pt-[O]_x-Bi structure totally different with PtBi alloy to improve the thermo-stability of catalysts and supply active oxygen species for efficiently catalyzing CO oxidation at low temperature.

3. *There are still some broken sentences. Revise the entire manuscript carefully.*

Author reply: we thank reviewer's comment. We have re-read entire manuscript carefully and amended the grammatical mistakes.

4. *Supplementary table 2 could be more informative. Please provide the calculated DEaverage and the structural models together.*

Author reply: we thank reviewer's suggestion. In the revised supplementary information, we have added the DFT models in supplementary table 2.

5. I read Supplementary Figure 9b differently from the authors. I see that the 1Pt-SiO₂-400 is less reactive for CO oxidation than 1Pt-SiO₂ at below 150 C. This could be originated from the following two cases: 1) different kinds of the reaction sites with different activation energy barriers are working in two systems or 2) 1Pt-SiO₂ provides dense active sites (sites/Pt used) than 1Pt-SiO₂-400. The authors may provide the site or mass normalized rate plots (TOF or TON).

Author reply: we thank reviewer's suggestion. We have acquired the kinetic data of 1Pt-SiO₂-400 with CO conversion < 20% and provided the mass normalized rate in Supplementary Table 4. The apparent activation energy (E_a) is 81 ± 5 kJ/mol, which is slightly higher than that (70 ± 4 kJ/mol) of 1Pt-SiO₂ (Fig. R1 and Supplementary Table 4). It indicates that there is a difference of active site for 1Pt-SiO₂ and 1Pt-SiO₂-400 in CO oxidation. According to the structural characterization results of used 1Pt-SiO₂ and 1Pt-SiO₂-400, metallic platinum clusters (3.0 ± 0.6 nm) and huge Pt particles (50-100 nm) coexist in the former and only platinum clusters (1.8 ± 0.1 nm) appear in the latter. As we all know, huge platinum particles make few contributions to CO oxidation activity. In addition, Casapu et al have reported that platinum nanoparticle size has huge influence on CO oxidation activity due to different CO adsorption strength and the ability to regenerate the active sites. The optimal Pt particle size of 2–3 nm was identified for CO oxidation on Pt/Al₂O₃ catalysts (Casapu et al. *ACS Catal.* 2017, 7, 343–355). Therefore, the existence of small-size platinum particles in used 1Pt-SiO₂-400 increases the number of active Pt surface sites. But, the in-situ DRIFTS results

show that the over strong CO adsorption (CO poison, Fig. R2) on platinum sites leads to not enough platinum sites to dissociate oxygen molecule, which requires higher temperature to motivate CO oxidation (Boubnov et al. *Appl. Catal., B* 2012, 126, 315–325; Boubnov et al. *Top. Catal.* 2013, 56, 333–338; Ono et al. *J. Phys. Chem. C* 2011, 115, 16856–16866.).

Fig. R1 Arrhenius plot for 1Pt-SiO₂-400.

Supplementary Table 4: Rates normalized by catalyst weight (r_w), rates normalized by platinum amount (r_{Pt}) and apparent activation energies (E_a) for the carbon monoxide oxidation reaction over Pt/PtBi-SiO₂ samples.

Sample	r_w ($\mu\text{mol}_{\text{CO}}\cdot\text{g}_{\text{cat}}^{-1}\cdot\text{s}^{-1}$)			r_{Pt} ($\text{mmol}_{\text{CO}}\cdot\text{mol}_{\text{Pt}}^{-1}\cdot\text{s}^{-1}$)			E_a (kJ/mol)
	90°C	100°C	110°C	90°C	100°C	110°C	
1Pt-SiO ₂	0.2	0.3	0.7	4.8	7.3	17.1	70±4
1Pt-SiO₂-400	0.2	0.3	0.4	4.8	7.3	9.8	81±5
1Pt2Bi-SiO ₂	1.7	3.1	3.9	41.4	75.5	95.1	52±4
1Pt5Bi-SiO ₂	0.9	1.5	2.2	21.9	36.6	53.6	56±3

Fig. R2 In-situ DRIFTS study of CO adsorption on 1Pt-SiO₂ and 1Pt-SiO₂-400.